# A male steroid controls female sexual behaviour in the malaria mosquito

Duo Peng[1], Evdoxia G. Kakani[1,5], Enzo Mameli[1,6], Charles Vidoudez[2], Sara N. Mitchell[1,5], Gennifer E. Merrihew[3], Michael J. MacCoss[3], Kelsey Adams[1], Tasneem A. Rinvee[1], W. Robert Shaw[1] & Flaminia Catteruccia[1,4 ✉]

Insects, unlike vertebrates, are widely believed to lack male-biased sex steroid hormones[1]. In the malaria mosquito *Anopheles gambiae*, the ecdysteroid 20-hydroxyecdysone (20E) appears to have evolved to both control egg development when synthesized by females[2] and to induce mating refractoriness when sexually transferred by males[3]. Because egg development and mating are essential reproductive traits, understanding how *Anopheles* females integrate these hormonal signals can spur the design of new malaria control programs. Here we reveal that these reproductive functions are regulated by distinct sex steroids through a sophisticated network of ecdysteroid-activating/inactivating enzymes. We identify a male-specific oxidized ecdysteroid, 3-dehydro-20E (3D20E), which safeguards paternity by turning off female sexual receptivity following its sexual transfer and activation by dephosphorylation. Notably, 3D20E transfer also induces expression of a reproductive gene that preserves egg development during *Plasmodium* infection, ensuring fitness of infected females. Female-derived 20E does not trigger sexual refractoriness but instead licenses oviposition in mated individuals once a 20E-inhibiting kinase is repressed. Identifying this male-specific insect steroid hormone and its roles in regulating female sexual receptivity, fertility and interactions with *Plasmodium* parasites suggests the possibility for reducing the reproductive success of malaria-transmitting mosquitoes.

Malaria cases and deaths are once more on the rise[4] owing to several factors including widespread insecticide resistance in *Anopheles* mosquitoes, which are the only vectors for human malaria parasites. The mating biology of these mosquitoes is a particularly attractive target for novel malaria control interventions because females mate only once[5]; rendering this single mating event sterile would have tremendous potential for reducing field mosquito populations.

Females lose their sexual receptivity after receiving high titres of steroid hormones from males. Studies have suggested that the trigger of refractoriness to further mating is 20-hydroxyecdysone (20E)[3], a steroid hormone better known as a regulator of molting cycles during larval stages[6]. The ability of males to synthesize and transfer 20E has evolved specifically in a subset of *Anopheles* species from the *Cellia* subgenus[7], which populates Africa and comprises the most dangerous malaria vectors, including *Anopheles gambiae*[5]. This is particularly notable because, in these species, 20E is also produced by females after every blood meal, whereby 20E drives the oogenetic cycles (reviewed in ref. [8]). However, the manner in which females integrate signals originating from two different ecdysteroid sources (male transferred and blood feeding induced) without impairing their own ability to mate is poorly understood. Indeed, if sexual refractoriness were to be triggered by female-produced 20E, this would cause sterility in individuals that

blood feed as virgins, which is a highly common behaviour among these mosquitoes[5].

A possible explanation is that *A. gambiae* males transfer a modified, male-specific ecdysteroid that activates signalling cascades in the female reproductive tract, leading to mating refractoriness. However, whereas vertebrates have multiple classes of largely dimorphic steroid hormones such as oestrogens and androgens (reviewed in ref. [9]), male-biased steroids have not been identified in insects, to the best of our knowledge.

## The ecdysteroid 20E is modified in males

We set out to determine the full composition of steroid hormones in the male accessory glands (MAGs) of sexually mature *A. gambiae* males, searching for possible modified steroids. Using high-performance liquid chromatography coupled with tandem mass spectrometry (HPLC–MS/MS) rather than the less-specific methods previously used[7,10,11], we detected ecdysone (E) and 20E in this tissue, confirming previous results. However, samples were dominated by an oxidized, phosphorylated steroid consistent with the chemical formula 3-dehydro-20E-22-phosphate (3D20E22P)[12] (Fig. 1). Other forms included 3-dehydro-20E (3D20E) and 20E-22-phosphate (20E22P).

[1]Department of Immunology and Infectious Diseases, Harvard T.H. Chan School of Public Health, Boston, MA, USA. [2]Harvard Center for Mass Spectrometry, Cambridge, MA, USA. [3]Department of Genome Sciences, University of Washington, Seattle, WA, USA. [4]Howard Hughes Medical Institute, Chevy Chase, MD, USA. [5]Present address: Verily Life Sciences, South San Francisco, CA, USA. [6]Present address: Department of Genetics, Blavatnik Institute, Harvard Medical School, Boston, MA, USA. ✉e-mail: fcatter@hsph.harvard.edu

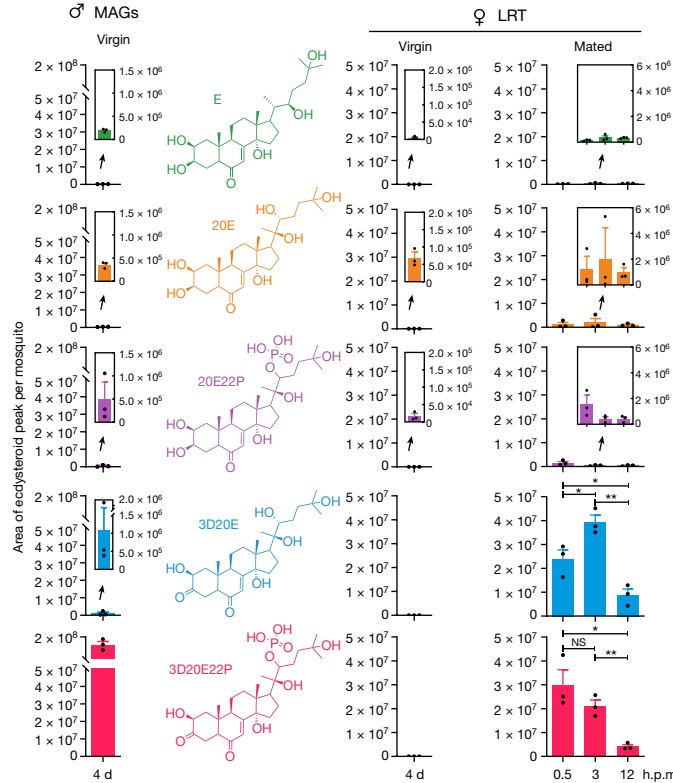

**Fig. 1 | Multiple ecdysteroids are produced in MAGs and transferred to the female LRT during mating.** MAGs and female LRTs (encompassing the atrium, spermatheca and parovarium) were dissected from 4-day-old (4 d) virgin males and from virgin and mated females (at 0.5, 3 and 12 h.p.m.). Ecdysteroids in these tissues were analysed by HPLC–MS/MS (mean ± s.e.m.; unpaired $t$-test, two-sided, false discovery rate (FDR)-corrected; NS, not significant; *$P < 0.05$, **$P < 0.01$. 3D20E: 3 h versus 0.5 h, $P = 0.035$; 12 h versus 3 h, $P = 0.0015$; 12 h versus 0.5 h, $P = 0.030$. 3D20E22P: 3 h versus 0.5 h, $P = 0.25$; 12 h versus 3 h, $P = 0.0032$; 12 h versus 0.5 h, $P = 0.015$). Data were pooled from three biological replicates. Peak area was calculated for each ecdysteroid of interest and normalized by mosquito numbers. Ecdysteroids are indicated by colour as follows: E, green; 20E, orange; 20E22P, purple; 3D20E, blue; 3D20E22P, pink. Insets increase the scale on the $y$-axis to show lower ecdysteroid levels.

The HPLC–MS/MS signal intensity of 3D20E22P was two orders of magnitude higher than that of its dephosphorylated form 3D20E and three orders of magnitude higher than that of E and 20E (Fig. 1). No detectable levels of 3D20E22P or 3D20E were found in blood-fed females, although E and 20E (and low levels of 20E22P) were detected in the rest of the body, as expected[11], and in the lower reproductive tract (LRT; Extended Data Fig. 1a). We also profiled ecdysteroids in newly eclosed (<1 day old) males and females and detected 3D20E and 3D20E22P only in the MAGs; E, 20E and 20E22P were present in both sexes (Extended Data Fig. 1b). These data demonstrate that *A. gambiae* adult males produce high titres of modified hormones in their MAGs that are not synthesized by females.

## 3D20E22P is dephosphorylated in the LRT

To investigate whether 3D20E22P and 3D20E are transferred during mating, we dissected the female LRT at different timepoints after mating. Whereas no ecdysteroid was identified in virgins, we observed a large amount of 3D20E22P in the LRT immediately following copulation (0.5 h post-mating, h.p.m.), which decreased over time paralleled by a significant increase in 3D20E levels (Fig. 1). Using chemically synthesized 3D20E as standard, we determined that levels of this steroid hormone in mated LRT were at least 100-fold higher than

those of 20E (Extended Data Table 1). 3D20E22P is therefore the main male ecdysteroid transferred to the female LRT during mating, and its dephosphorylated form 3D20E becomes highly abundant shortly after mating. This suggests a prominent role of the latter ecdysteroid in female post-mating biology.

## Identification of 20E-modifying enzymes

Using a custom-built bioinformatics pipeline after generation of new RNA-sequencing (RNA-seq) datasets (Fig. 2a), we searched for genes encoding 20E-modifying ecdysteroid kinases (EcK), ecdysone oxidases (EO) and ecdysteroid phosphate phosphatases (EPP) expressed in reproductive tissues. We identified one candidate EPP gene and two potential EcK genes (*EcK1* and *EcK2*), but were unable to find a good candidate EO gene. Notably, the single EPP gene was expressed at high levels (98.9th percentile) in *A. gambiae* MAGs but not in the female LRT (Fig. 2b), contrary to our expectations given that dephosphorylation of 3D20E22P occurs in this female tissue. We therefore considered that male EPP may be transferred during copulation. Indeed, we identified this enzyme by MS in the atrium of females after mating using in vivo stable isotope labelling to mask female proteins (Fig. 2c and Supplementary Table 1). EPP presence in the MAGs and in mated (but not virgin) female LRT was also confirmed using a specific antibody (Fig. 2d).

The ecdysteroid phosphate phosphatase activity of EPP was validated after incubation with 3D20E22P isolated from the MAGs by HPLC–MS/MS (Extended Data Fig. 2a). Moreover, when we silenced *EPP* by RNA-mediated interference (RNAi), we detected a strong decrease in phosphatase activity in these male reproductive tissues (Fig. 3a), and females mated to *EPP*-silenced males showed a significantly lower proportion of dephosphorylated 3D20E (Fig. 3b) despite partial gene silencing (Extended Data Fig. 2b,c). Conversely, we did not detect a significant change in the 20E22P/20E ratio in the same mosquitoes, possibly suggesting specificity of this enzyme for 3D20E22P (Fig. 3b).

## 3D20E prevents female remating

We next evaluated whether ecdysteroid dephosphorylation is important for inducing female refractoriness to mating. Remarkably, females mated to *EPP*-depleted males remated at substantially higher frequencies (44.9%) than control females (10.4%) when exposed to additional (transgenic) males (Fig. 3c). We also observed a significant decrease in fertility (Fig. 3d, left) and a marginal decrease in the number of eggs laid by these females (Fig. 3d, centre), whereas the percentage of females ovipositing (another response triggered in females by mating) was not affected (Fig. 3d, right). Given the observed specificity of EPP for 3D20E22P, these results suggest that the activation of 3D20E by EPP transferred during mating may have a prominent role in switching off female receptivity to further copulation, a behaviour that has been previously attributed to sexual transfer of 20E. Hence, this male-specific hormone also strongly affects female fertility.

## 3D20E is more active than 20E

We next compared the activity of 20E and 3D20E in injection experiments in sexually mature virgin females, using chemically synthesized 3D20E (Fig. 4a–c) and commercially available 20E. We observed that 3D20E was significantly more effective than 20E in turning off the female's susceptibility to mating at two concentrations (Fig. 4d). Notably, 24 h after injections at the highest concentration, half of the physiological level of 3D20E in the LRT (1,066 pg after injections versus 2,022 pg after mating) induced a greater proportion of refractory females than 20 times the physiological level of 20E (361 pg after injections versus 18 pg after mating; Extended Data Table 1). This result is in agreement with the notion that sexual transfer of 20E does not induce mating refractoriness and further points to 3D20E as the principal factor

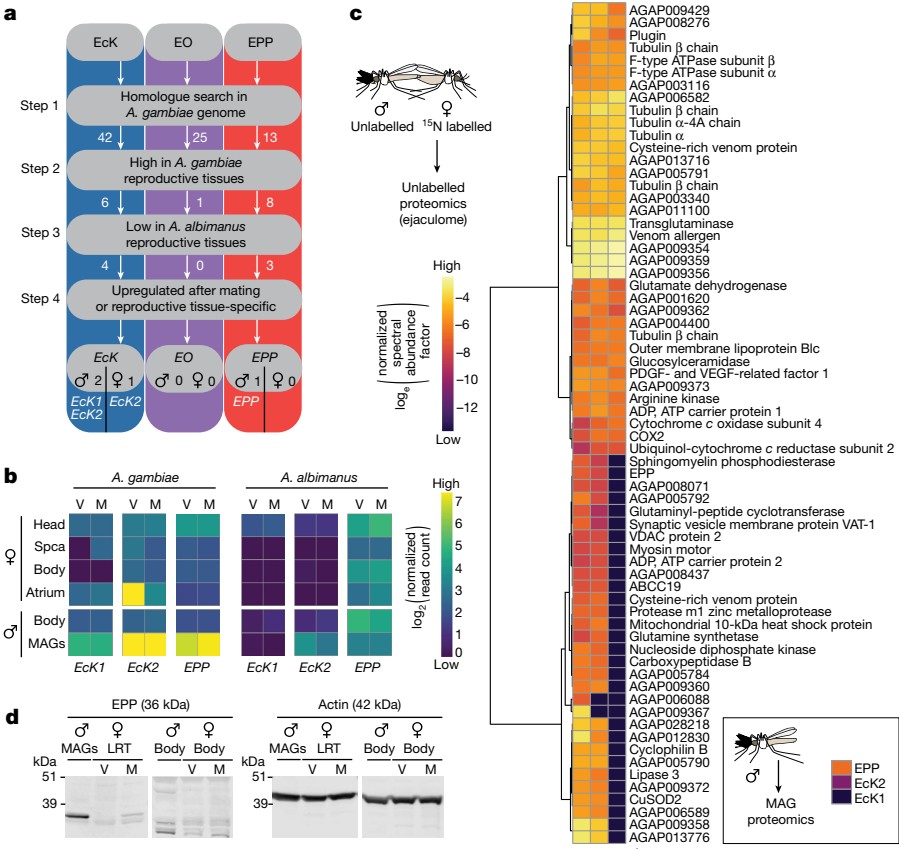

**Fig. 2 | Identification of two EcK genes and one EPP gene in mosquito reproductive tissues. a**, The custom-built bioinformatic pipeline used to search for genes encoding EcKs, EOs and EPPs in the reproductive tissues of each sex. Numbers next to arrows indicate the number of male and female candidates at each step. This analysis identified one EPP gene (*EPP*) and one EcK gene (*EcK1*) expressed in males, and one EcK gene (*EcK2*) expressed in both sexes but did not yield a candidate EO gene. **b**, Heat map comparing expression of candidate genes in tissues of virgin (V) and mated (M) *A. gambiae* and *Anopheles albimanus*. Spca, spermatheca; MAGs, male accessory glands; body, the rest of the body including the thorax, wings, legs, fatbody and guts in both sexes and ovaries in females. *EcK2* was highly expressed in both the MAGs and atria of *A. gambiae*, whereas *EPP* was found only in the MAGs. **c**, Proteomics

analysis of the male ejaculome transferred to the female atrium at 3, 12 and 24 h.p.m., showing the 67 most abundant proteins. Females were reared with food containing [15]N to label (and mask) all proteins. Unlabelled males were mated with labelled females, and the female LRTs were dissected at 3, 12 and 24 h.p.m. for proteomic analysis (see Supplementary Table 1 for a complete list of ejaculome proteins). Inset, EPP, Eck1 and EcK2 detected in the MAGs of virgin males by proteomic analysis of these tissues. **d**, EPP detected by western blot in the MAGs and mated female LRTs but not in virgin females or in the rest of the body of either males or females. The membrane was probed simultaneously with anti-actin (loading control) and anti-EPP antibodies. All males were virgins. For gel source data, see Supplementary Fig. 1. The western blot was performed twice with similar results.

ensuring paternity. 3D20E also showed significantly higher activity than 20E in oviposition assays in virgin females (Fig. 4e), which suggests that the normal oviposition rates we observed after partial *EPP* silencing were due to female factors still triggered by mating in the presence of residual 3D20E activity.

In previous studies, we determined that sexual transfer of steroid hormones induces the expression of *MISO* (mating-induced stimulator of oogenesis[11]), a female reproductive gene that protects *A. gambiae* females from the fitness costs otherwise inflicted by infection with *Plasmodium falciparum*[13], the deadliest human malaria parasite. Given the importance of *MISO* for *Anopheles* reproductive fitness in malaria-endemic regions, we decided to determine which hormone, 3D20E or 20E, triggers the expression of this gene. We found that whereas 20E injections specifically induced or more potently induced some nuclear hormone receptors (HRs) such as *HR3* and *HR4* and canonical downstream steroid targets such as the vitellogenesis gene *Vg*[14–16], *MISO* was more strongly induced by 3D20E (Extended Data Fig. 3). Sexual transfer of this male steroid hormone therefore appears to induce mechanisms that protect females from costs otherwise inflicted by parasite infection. Moreover, 3D20E differentially affected two isoforms of the E receptor *EcR*,

inducing *EcR-A* and repressing *EcR-B*, and more strongly triggered other mating-induced genes including *HPX15*, which affects female fertility[17]. This might explain the significant infertility observed in females mated to *EPP*-silenced males (Extended Data Fig. 3). These data suggest the existence of downstream pathways preferentially activated by the two ecdysteroids that may specifically underlie sex-specific functions.

## *EcK2* functions in males and females

We next tested the function of the two EcK genes identified in our bioinformatics pipeline. Silencing either *EcK1* or *EcK2* induced significant mortality in males (Extended Data Fig. 4a), showing that ecdysteroid phosphorylation, and thus deactivation[12], is important for survival. Because *EcK2* is expressed at higher levels than *EcK1* and was detected in MAGs by proteomics (Fig. 2b,c and Supplementary Table 2), we validated its ecdysteroid kinase activity by incubating it with 20E, which resulted in phosphorylated 20E22P (Extended Data Fig. 4b). When 3D20E was used instead as the substrate, we could not detect the phosphorylation product 3D20E22P (Extended Data Fig. 4c), indicating that 20E rather than 3D20E may be a preferred target of EcK2.

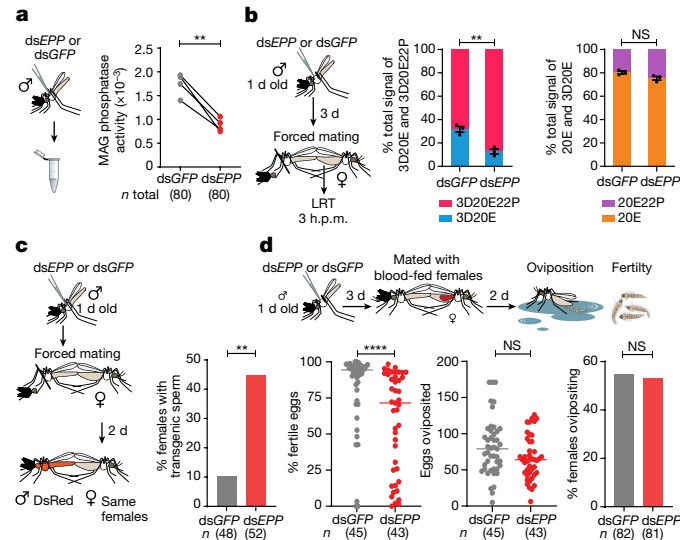

**Fig. 3 | Sexually transferred EPP regulates female remating rates and fertility. a**, Decreased phosphatase activity in MAGs caused by *EPP* silencing using double-stranded EPP RNA (ds*EPP*) or double-stranded GFP RNA (ds*GFP*) control. A pool of 20 MAGs was used in each replicate ($P = 0.0046$, paired *t*-test, two-sided), indicated by separate dots. **b**, Females mated to *EPP*-silenced males have a significantly lower proportion of dephosphorylated 3D20E at 3 h.p.m. ($P = 0.0043$, unpaired *t*-test, two-sided), whereas 20E levels are unaffected ($P = 0.063$, unpaired *t*-test, two-sided). The data are presented as mean ± s.e.m., derived from three pools of 13, 16 and 19 females each. **c**, Females mated to *EPP*-silenced males have a significantly higher rate of remating ($P = 0.0002$, Fisher's exact test, two-sided). The females were first force-mated to ensure their mated status; 2 days later, they were exposed to additional males carrying transgenic sperm to assess remating rates by quantitative PCR detection of the transgene. **d**, Blood-fed females mated to *EPP*-silenced males have a significant decrease in fertility ($P < 0.0001$; Mann–Whitney test, two-sided) and a slight decrease in egg numbers ($P = 0.088$, Mann–Whitney test, two-sided), whereas the oviposition rate is not affected ($P = 0.94$, Fisher's exact test, two-sided). In all panels, *n* indicates the number of biologically independent mosquito samples. NS, not significant. *$P < 0.05$, **$P < 0.01$, ***$P < 0.001$, ****$P < 0.001$.

According to our RNA-seq analysis, *EcK2* is also highly expressed in the LRT of virgin females, where it is turned off after mating (Fig. 2b). We confirmed these data and determined that *EcK2* expression is not affected by blood feeding (Extended Data Fig. 5a). Expanding our initial MS experiments, we determined that the peak of 20E22P closely mirrored that of 20E (22–26 h after a blood meal; Extended Data Fig. 5b). Silencing of *EcK2* in virgin females caused a 3-fold increase in the relative ratio of 20E to 20E22P 26 h after a blood meal (Extended Data Figs. 2c and 5c), confirming that EcK2 also phosphorylates 20E in females. Notably, *EcK2*-depleted virgins maintained full sexual receptivity (Extended Data Fig. 5d,e), which further demonstrates that female-produced 20E does not induce mating refractoriness. However, these females showed a significant increase in oviposition rates relative to controls, with more than 30% of virgins ovipositing eggs (Extended Data Fig. 5f). Oviposition did not occur if double-stranded *Eck2* RNA (ds*EcK2*) injections were carried out after blood feeding, when the 20E peak induced by blood ingestion had already declined. Overall, these results support a model whereby 20E produced after blood-feeding can induce oviposition but only when oviposition blocks (EcK2 and possibly other factors) are turned off by mating. Neither 20E nor 3D20E injections suppressed *EcK2* expression in virgins (Extended Data Fig. 5g), which indicates that other factors mediate repression of this kinase. 20E levels after blood feeding, however, are not sufficient to induce refractoriness to mating, which is instead efficiently triggered by high titres of sexually transferred 3D20E.

## Discussion

Our results provide critical insight into the mechanisms regulating *A. gambiae* reproductive success. A model emerges whereby males have evolved to synthesize high titres of 3D20E, a male-specific, modified ecdysteroid that warrants paternity by specifically desensitizing females to further mating. In parallel, these malaria vectors have also evolved an efficient system to activate 3D20E in females based on the sexual transfer of male-specific EPP. To our knowledge, this is the first example of a system of male- and female-predominant steroid hormones exerting distinct and key functions in insects. The existence of male-specific ecdysteroid functions has been postulated but not

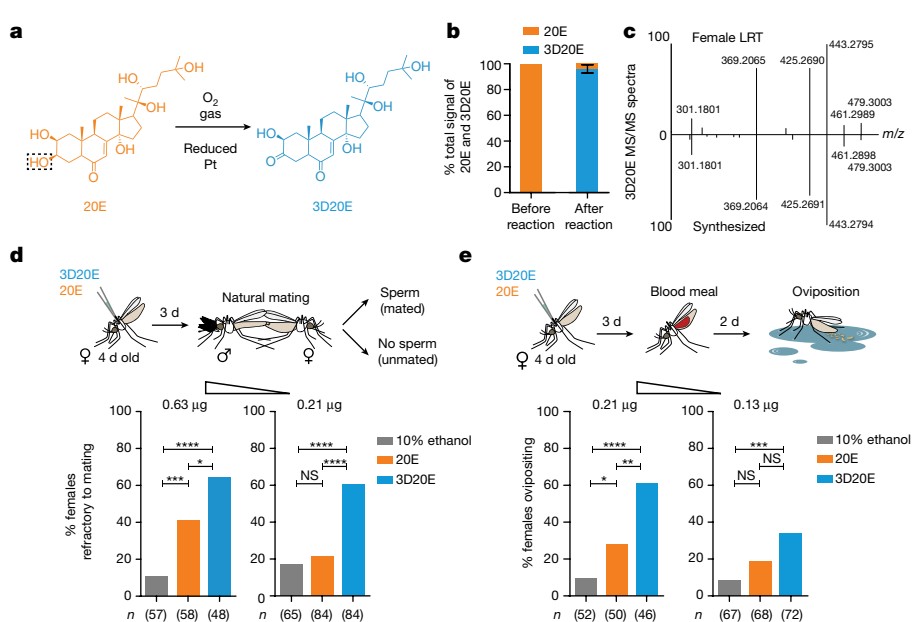

**Fig. 4 | 3D20E is more potent than 20E at inducing mating refractoriness and oviposition. (a,b)** 3D20E chemically synthesized from 20E (**a**) with very high levels of conversion/efficiency (data presented as mean ± s.e.m., derived from three independent synthesis reactions) (**b**). **c**, Mass spectra (lower half) perfectly matched those of ecdysteroids found in the mated female LRT (upper half). **d**, Injection of 0.63 μg or 0.21 μg 3D20E induced significantly higher refractoriness to mating than 20E (0.63 μg, $P = 0.02$; 0.21 μg, $P < 0.0001$; Fisher's exact tests, two-sided) and 10% ethanol controls (0.63 μg, $P < 0.0001$; 0.21 μg, $P < 0.0001$; Fisher's exact tests, two-sided), whereas 20E was significantly higher than controls only at the higher dose (0.63 μg, $P = 0.0002$; 0.21 μg, $P = 0.54$; Fisher's exact tests, two-sided). **e**, 3D20E injections induced significantly higher oviposition rates than 10% ethanol controls in virgin females (0.21 μg, $P < 0.0001$; 0.13 μg, $P = 0.0003$; Fisher's exact tests, two-sided), whereas 20E was significant compared with controls only at the higher dose (0.21 μg, $P = 0.022$; 0.13 μg, $P = 0.0823$; Fisher's exact tests, two-sided). 3D20E induced significantly higher oviposition rates than 20E at higher doses (0.21 μg, $P = 0.0019$; 0.13 μg, $P = 0.075$; Fisher's exact tests, two-sided). In all panels, *n* indicates the number of biologically independent mosquito samples. NS, not significant. *$P < 0.05$, **$P < 0.01$, ***$P < 0.001$, ****$P < 0.001$. Data were pooled from three replicates.

definitively demonstrated. For example, a largely disproved hypothesis[18] is that such functions may be carried out by the 20E precursor E[1]. It is well established that, in *Drosophila*, monandry is instead triggered by the sexual transfer of small sex peptides[19,20] that interact with neurons innervating the female reproductive tract through specific sex peptide receptors[21,22]. Further work is needed to identify the downstream signalling cascades controlled by 3D20E in *A. gambiae* females and to determine whether these cascades may be conserved between mosquitoes and fruit flies.

Given the important role of 3D20E for female fertility and behaviour determined in our study, the pathways leading to 3D20E synthesis and activation provide novel opportunities for future mosquito control strategies, such as generating competitive sterile males for field releases in sterile insect technique strategies[23] or mimicking 3D20E function in virgin females. A male-specific function for 3D20E may have evolved when *A. gambiae* and other *Cellia* species acquired the ability to coagulate their seminal fluid into a mating plug[7] because this allowed for effective transfer of large amounts of hormones and hormone-activating enzymes. In turn, the evolution of 3D20E to enforce monandry also equipped females with mechanisms (through the high expression of *MISO*) favouring their reproductive fitness in areas of high malaria endemicity, which indirectly promotes the spread of *Plasmodium* parasites. Given that female 20E has been shown to profoundly affect the survival and growth of *P. falciparum* in *Anopheles* females[24], both male and female steroid hormone pathways are now key facets in mosquito–parasite interactions.

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

## Methods

### Rearing of *A. gambiae* mosquitoes

The *A. gambiae* G3 strain was reared under standard insectary conditions (26–28 °C, 65–80% relative humidity, 12:12 h light/darkness photoperiod). Larvae were fed on powdered fish food (TetraMin Tropical Flakes, Koi Pellets and Tetra Pond Sticks in a 7:7:2 ratio). Adult mosquitoes were fed ad libitum on 10% glucose solution and weekly on human blood (Research Blood Components). Virgin mosquitoes were obtained by separating the sexes at the pupal stage after microscopic examination of the terminalia. Males carrying the *DsRed* transgene have been previously described[25].

### Forced and natural mating

Forced mating experiments were conducted according to protocols described previously[13]. For natural mating, 4-day-old virgin females were kept for two nights with sexually mature virgin males at a 1:3 ratio. For experiments using ds*EPP*-injected males, co-caging coincided with days 3–4 after injection, when phosphatase activity was maximally silenced (Extended Data Fig. 2b).

### HPLC–MS/MS sample preparation for ecdysteroids

Mosquito tissues, remaining carcasses (rest of body) or whole bodies were dissected into 100% methanol and homogenized with a bead beater (2-mm glass beads, 2,400 r.p.m., 90 s). The number of tissues and volume of methanol were as follows: rest of body, 50 in 1,000 µl; MAGs, 50–100 in 80 µl; female LRT, 25–50 in 80 µl. Methanol extraction was performed a second time on the pellet using the same volume of methanol. Cellular debris was removed by centrifugation. Methanol from two extractions was combined and dried under nitrogen flow and resuspended in the following volumes of 80% methanol in water: rest of body, 50 µl; MAGs and female LRT, 30 µl.

### HPLC–MS/MS analysis

Samples were analysed on a mass spectrometer (ID-X, Thermo Fisher) coupled to an LC instrument (Vanquish, Thermo Fisher). Five microlitres of sample was injected onto a 3-µm, 100 × 4.6 mm column (Inspire C8, Dikma) maintained at 25 °C. The mobile phases for the LC were A (water, 0.1% formic acid) and B (acetonitrile, 0.1% formic acid). The LC gradient was as follows: 1 min at 5% B followed by an increase to 100% B in 11 min. After 8 min at 100%, the column was re-equilibrated at 5% B for 4 min. The flow rate was 0.3 ml min$^{-1}$. Ionization in the MS source was done by heated electrospray ionization in positive and negative modes.

The mass spectrometer measured data in full-MS mode at 60,000 resolution over an *m/z* range of 350 to 680. The MS/MS data were acquired on $[M + H]^+$ (all targets), $[M - H_2O + H]^+$ (all targets) and $[M - H]^-$ (phosphorylated targets). The MS/MS data were used to confirm the ecdysteroid nature of the targets for which no standards were available. To identify untargeted ecdysteroids, the MS/MS data for all HPLC peaks of >15% relative abundance were analysed. Quantification was done using standard curves created from either pure standards (20E, 3D20E) to calculate absolute amounts or dilutions of one specific sample (all other targets) to calculate their equivalence to the amount found in one male. For 3D20E, quantification was performed with the sum of the following adducts: $[M + TFA]^-$, $[M + COOH]^-$, $[M + Na]^+$, $[M + Cl]^-$, $[M + NO_3]^-$. Data were extracted and quantified using Tracefinder (version 4.1). MS/MS data were analysed using Xcalibur (version 4.4). The MS spectra of E, 20E and 3D20E were compared to respective standards. 3D20E22P was analysed by derivatization with Girard's reagent. 20E22P was analysed by *m/z* ratio.

### HPLC–MS/MS purification of 3D20E22P

3D20E22P was purified from MAGs. The purification was performed at analytical scale under the same LC conditions as for HPLC–MS/MS analysis using an ultra-performance liquid chromatography instrument (Acquity, Waters) coupled with a quadrupole mass-based detector (QDa, Acquity, Waters). Fraction collection was triggered when the *m/z* corresponding to 3D20E22P was detected at the same retention time as that previously identified. The purity of the extracted compounds was then checked by HPLC–MS/MS as described above.

### RNA extraction, cDNA synthesis and RT–qPCR

Total RNA was extracted with TRI reagent (Thermo Fisher) from pools of 10–12 reproductive tissues or the rest of the body (headless) following the manufacturer's instructions. RNA was treated with TURBO DNase (Thermo Fisher). cDNA was synthesized using Moloney murine leukaemia virus reverse transcriptase (M-MLV RT; Thermo Fisher) following the manufacturer's instructions. Primers used for quantitative PCR with reverse transcription (RT–qPCR; Extended Data Table 2) were previously published[24] or were designed using Primer-BLAST[26] with preferences for products 70–150 bp in size and for primers spanning exon–exon junctions or primer pairs on separate exons. cDNA samples from three to four biological replicates were diluted four-fold in water for RT–qPCR. Quantification was performed in 15-µl duplicated reactions containing 1× PowerUp SYBR Green Master Mix (Thermo Fisher), primers and 5 µl of diluted cDNA. Reactions were run on a QuantStudio 6 Pro Real-Time PCR System (Thermo Fisher), and the data were collected and analysed using Design and Analysis (version 2.4.3). Relative quantities were normalized against the ribosomal gene *RpL19* (AGAP004422), the expression of which does not change significantly with blood feeding[27] or mating[3], as confirmed in this study.

### RNA-seq analysis

RNA quality was checked with an Agilent bioanalyser 2100 Bioanalyzer (Agilent). Illumina paired-end libraries were prepared and run at the Broad Institute of MIT and Harvard. Sequencing reads were aligned to the *A. gambiae* genome (PEST strain, version 4.12) using HISAT2 (version 2.0.5) with the default parameters. Reads with mapping quality (MAPQ) scores <30 were removed using Samtools (version 1.3.1). The numbers of reads mapped to genes were counted using htseq-count (version 0.9.1) with the default parameters. Calculation of normalized read counts and analysis of differential gene expression was performed using the DESeq2 package (version 1.28.1) in R (version 4.0.3).

### Bioinformatic pipeline

Ecdysteroid-modifying gene candidates were identified by first searching the *A. gambiae* genome with the PSI-BLAST algorithm (https://ftp.ncbi.nlm.nih.gov/blast/executables/blast+/2.8.1/) using the default parameters with the following query protein sequences: EcK from *Bombyx mori* (accession NP_001038956.1), *Musca domestica* (accessions XP_005182020.1, XP_005175332.1 and XP_011294434.1) and *Microplitis demolitor* (accessions XP_008552646.1 and XP_008552645.1); EPP from *B. mori* (accession NP_001036900), *Drosophila melanogaster* (accession NP_651202), *Apis mellifera* (accession XP_394838) and *Acyrthosiphon pisum* (accession: XP_001947166); and EO from *B. mori* (accessions NP_001177919.1 and NP_001243996.1) and *D. melanogaster* (accession NP_572986.1) (step 1). Next, hit genes were filtered on the basis of high mRNA expression (>100 fragments per kilobase of exon per million mapped reads (FPKM) or >85th percentile) in reproductive tissues (female LRT or MAGs) of *A. gambiae* (step 2). For improved specificity, we selected against candidate enzymes also expressed in the reproductive tissues of *A. albimanus*, an anopheline species that does not synthesize or transfer ecdysteroids during mating[7]; candidate genes were filtered on the basis of low expression (<100 FPKM or <85th percentile) in reproductive tissues of *A. albimanus* (step 3). As a final filter (step 4), candidate genes needed to satisfy at least one of the following: (1) significant upregulation after mating (*P* < 0.05) according to the analysis of differentially expressed genes and (2) lack of expression in non-reproductive tissues (<85th percentile or <100 FPKM).

## Whole-female stable isotope labelling

We modified a previously described method[28–30] to achieve whole-organism isotopic labelling. In brief, wild-type *Saccharomyces cerevisiae* type II (YSC2, Sigma) was grown in medium containing (wt/vol) 2% glucose (G7528, Sigma), 1.7% yeast nitrogen base without amino acids and ammonium sulfate (BD Difco, DF0335) and 5% $^{15}N$ ammonium sulfate (NLM-713, >99%, Cambridge Isotope Laboratories) as the only source of nitrogen. Yeast was recovered by centrifugation and fed ad libitum to mosquito larvae until pupation. Powdered fish food was supplemented (0.5 mg per 300 larvae) to prevent death at the fourth instar stage. Only females were then used in mating experiments with unlabelled males to analyse the male proteome transferred during copulation.

## Male ejaculate proteome (ejaculome)

$^{15}N$-labelled virgin females of 4–6 days old were force-mated to age-matched unlabelled virgin males. Successful mating was verified by detection of a mating plug under an epifluorescence microscope. At 3, 12 and 24 h.p.m., the atria of 45–55 mated females were dissected into 50 µl of ammonium bicarbonate buffer (pH 7.8) and homogenized with a pestle. The homogenate was centrifuged, and the supernatant was combined with 50 µl of 0.1% RapiGest (186001860, Waters) in 50 mM ammonium bicarbonate. The supernatant and pellet from each sample were snap-frozen on dry ice and shipped overnight to the Mac-Coss Lab at the University of Washington, where sample preparation for LC–MS/MS was completed. The pellets were resuspended in 50 µl of 0.1% RapiGest in 50 mM ammonium bicarbonate and sonicated in a water bath. Protein concentration was measured for both pellets and supernatants by BCA assay, and the samples were reduced with 5 mM dithiothreitol (DTT; Sigma) alkylated with 15 mM iodoacetamide (Sigma) and digested with trypsin for 1 h at 37 °C (1:50 trypsin:substrate ratio). RapiGest was cleaved with the addition of 200 mM HCl followed by 45 min of incubation at 37 °C and centrifugation at 14,000 r.p.m. for 10 min at 4 °C to remove debris. Samples were cleaned by dual-mode solid-phase extraction (Oasis MCX cartridges, Waters) and resuspended in 0.1% formic acid at a final protein concentration of 0.33 µg µl$^{-1}$. Unlabelled MAG proteomes were similarly analysed from virgin males. Two analytical replicates were analysed for each sample. Next, 1 µg of each sample digest was analysed using a 25-cm fused silica 75-µm column and a 4-cm fused silica Kasil1 (PQ) frit trap loaded with Jupiter C12 reverse phase resin (Phenomenex) with a 180-min LC–MS/MS run on a Q-Exactive HF mass spectrometer (Thermo Fisher) coupled with a nanoACQUITY UPLC system (Waters). Data-dependent acquisition data generated from each run were converted to mzML format using Proteowizard (version 3.0.20287) and were searched using Comet[31] (version 3.2) against a FASTA database containing protein sequences from *A. gambiae* (VectorBase release 54), *Anopheles coluzzi* Mali-NIH (VectorBase release 54), *S. cerevisiae* (Uniprot, March 2021), three-frame translations of *A. gambiae* RNA-seq and known human contaminants. Peptide-spectrum match FDRs were determined using Percolator[32] (version 3.05) at a threshold of 0.01, and peptides were assembled into protein identifications using protein parsimony in Limelight[33] (version 2.2.0). Relative protein abundance was estimated using a normalized spectral abundance factor (NSAF) calculated for each protein within each run as previously described[28]. The NSAF relative to each protein was averaged across samples from two different biological replicates. $^{15}N$ labelling successfully masked the female proteome, although a small number of unlabelled proteins could be detected from labelled virgins. We documented the detection at reduced amounts (1–5 spectra) of male proteins in female virgin samples only in technical runs in which the virgin samples were run after male/mated samples, as a result of HPLC 'carryover'. Proteins that were occasionally found as 'contaminants' from labelled virgins are listed in Supplementary Table 1.

## Customized antibody and western blotting

Two antigenic peptides, QTTDRVAPAPDQQQ (within isoform PA) and MESDGTTPSGDSEQ (within both isoforms PA and PB), from the EPP protein sequence were identified on the basis of predictions of antigenicity, surface exposure probability and hydrophilicity provided as a part of the customized antibody service at Genscript. The two peptides were pooled and were then conjugated with carrier protein KLH and injected into New Zealand rabbits. The rabbits were killed after the fourth injection, and total IgG was isolated with affinity purification. IgG from the most EPP-specific rabbit was used in further western blotting.

For western blotting, MAGs (*n* = 10, where *n* indicates the number of biologically independent mosquito samples) and female LRTs (*n* = 30) were dissected from 4-day-old virgin males and virgin or force-mated females (<10 min after mating), respectively, into protein extraction buffer (50 mM Tris, pH 8.0; 1% NP-40; 0.25% sodium deoxycholate; 150 mM NaCl; 1 mM EDTA; 1× protease inhibitor mixture (Roche)). The samples were immediately homogenized after dissection with a bead beater (2-mm glass beads, 2,400 r.p.m., 90 s). The insoluble debris was removed by centrifugation at 20,000*g* at 4 °C. Protein was quantified by Bradford assay (Bio-Rad). Then, 20 µg of MAG protein, 40 µg of LRT protein and 20 µg of rest of body protein were denatured and separated by 10% Bis–Tris NuPAGE using MOPS buffer. Protein was transferred to a polyvinylidene difluoride membrane using an iBlot2 transfer system (Thermo Fisher). The membranes were washed twice in 1× PBS-T (0.1% Tween-20 in PBS) and were then blocked for 1 h at 22 °C in Odyssey Blocking Buffer (Li-Cor). The membranes were incubated with the custom rabbit anti-EPP polyclonal primary antibody (1:700 in blocking buffer) and rat anti-actin monoclonal primary antibody MAC237 (Abcam; 1:4,000) with shaking overnight at 4 °C. The membranes were washed with PBS-T and were then incubated with secondary antibodies (donkey anti-rabbit 800CW and goat anti-rat 680LT (Li-Cor), both at 1:20,000) in blocking buffer with 0.01% SDS and 0.2 % Tween-20 for 1 h at 22 °C. The membranes were washed with PBS-T and imaged with an Odyssey CLx scanner. Images were collected and processed in Image Studio (version 5.2). No specific band corresponding to the EPP-RA isoform (82 kDa) was detected.

## Recombinant protein and enzymatic assays

The coding regions of *EPP* (as isoform AGAP002463-RB containing the histidine phosphatase domain, NCBI Conserved Domain Search[34]) and *EcK2* (AGAP002181) were cloned into the pET-21a(+) plasmid (Novagen Millipore Sigma); primers are listed in Extended Data Table 2. Eight GS4 linkers (in tandem) were inserted before the C-terminal 6×His tag in the pET-21a(+)-*EcK2* construct. Recombinant proteins were produced with NEBExpress cell-free *Escherichia coli* protein synthesis reactions (New England BioLabs). The recombinant proteins were purified with NEBExpress Ni spin columns (New England BioLabs). The dihydrofolate reductase (DHFR) control protein was produced using the DNA template from the NEBExpress cell-free *E. coli* protein synthesis kit. Proteins were stored in 50% glycerol in PBS at −20 °C for up to 3 months.

The phosphatase activity of EPP and tissue extracts was measured using 4-nitrophenyl phosphate (pNPP; Sigma-Aldrich). The reaction buffer contained 25 mM Tris, 50 mM acetic acid, 25 mM Bis–Tris, 150 mM NaCl, 0.1 mM EDTA and 1 mM DTT. Tissues were homogenized in reaction buffer, and cellular debris was removed by centrifugation. The reaction was initiated by adding enzyme or tissue extract to the reaction buffer containing 2.5 mg ml$^{-1}$ pNPP. The reaction mixture was incubated at room temperature in darkness, and the amount of pNP converted from pNPP was quantified by measuring the absorption at 405 nm at various times.

For in vitro EcK activity, proteins were incubated with 0.2 mg of 20E or 3D20E in 200 µl of buffer (pH 7.5) containing 10 mM HEPES–NaOH, 0.1% BSA, 2 mM ATP and 10 mM MgCl$_2$ at 27 °C for 2 h. The reaction was terminated by adding 800 µl of methanol and was then chilled for 1 h

at −20 °C followed by centrifugation at 20,000$g$ for 10 min at 4 °C. The supernatant was then analysed by HPLC–MS/MS. To heat-inactivate proteins used in the control groups, the proteins were incubated at 95 °C for 20 min in 50% glycerol in PBS.

For in vitro EPP activity, proteins were incubated with 3D20E22P (equivalent to the amount found in 18 pairs of MAGs, purified by HPLC–MS/MS) in 100 μl of buffer (pH 7.5) containing 25 mM Tris, 50 mM acetic acid, 25 mM Bis–Tris, 150 mM NaCl, 0.1 mM EDTA and 1 mM DTT at 27 °C for 3 h. The reaction was terminated by adding 400 μl of methanol and was chilled for 1 h at −20 °C followed by centrifugation at 20,000$g$ for 10 min at 4 °C. The supernatant was analysed by HPLC–MS/MS.

### dsRNA production
PCR fragments of *EPP* (362 bp), *EcK1* (AGAP004574, 365 bp) and *EcK2* (556 bp) were amplified from cDNA prepared from mixed-sex headless mosquito carcasses. A PCR fragment of the *eGFP* control (495 bp) was amplified from pCR2.1-*eGFP* described previously[11]; the PCR primers are listed in Extended Data Table 2. PCR fragments were inserted between the inverted T7 promoters on the pL4440 plasmid. Plasmid constructs were recovered from NEB 5-alpha Competent *E. coli* (New England Biolabs) and were verified by DNA sequencing before use (see Supplementary Data 1 for insert sequences). A primer matching the T7 promoter (Extended Data Table 2) was used to amplify inserts from pL4440-based plasmids. PCR product sizes were confirmed by agarose gel electrophoresis. dsRNA was transcribed from the PCR templates using the Megascript T7 transcription kit (Thermo Fisher) and was purified following the manufacturer's instructions with modifications previously described[35].

### Thorax microinjection
For dsRNA injection, 1,380 ng of dsRNA (ds*GFP*, ds*EcK1*, ds*EcK2*, ds*EPP*) was injected (Nanoject III, Drummond) at a concentration of 10 ng nl$^{-1}$ into the thorax of adult males or females within 1 day of eclosion. Gene knockdown levels were determined in at least three biological replicates by RNA extraction, cDNA synthesis and RT–qPCR. For ecdysteroid injections, depending on the experimental design, 4-day-old virgin or 6-day-old virgin blood-fed females were injected (Nanoject III, Drummond) with 0.13, 0.21 or 0.63 μg of 20E or 3D20E at a concentration of 1.3, 2.1 or 6.3 ng nl$^{-1}$, respectively. Ten percent (vol/vol) ethanol in water was injected at a volume of 100 nl; 3D20E22P in 10% ethanol was injected at a volume of 100 nl (equivalent to 75% of the amount found in a pair of MAGs). Mosquitoes were randomly assigned to injection groups.

### Oviposition and remating assays
For oviposition assays, 3-day-old females were blood-fed ad libitum on human blood. Partially fed or unfed mosquitoes were removed. Depending on the treatment, after a minimum of 48 h after the blood meal, females were placed into individual oviposition cups for four nights. Eggs were counted under a stereoscope (Stemi 508, Zeiss); for mated females, eggs that hatched into larvae were scored as fertile.

For mating assays, depending on the treatment, females were allowed a minimum of 2 days to develop refractoriness to mating, and wild-type age-matched males were subsequently introduced to the same cage. After two nights, spermathecae were dissected from females, and the genomic DNA was released by freeze–thawing and sonication in a buffer containing 10 mM Tris–HCl, 1 mM EDTA and 25 mM NaCl (pH 8.2). The samples were incubated with proteinase K (0.86 μg μl$^{-1}$) for 15 min at 55 °C followed by 10 min at 95 °C. The crude genomic DNA preparation was diluted 10-fold and was subjected to qPCR detection of a Y-chromosome sequence; the primers are listed in Extended Data Table 2. The absence of a Y-chromosome sequence is indicative of no mating.

For remating assays, force-mated females were checked for the presence of mating plugs to confirm mating status and were allowed 2 days to develop refractoriness to mating in the absence of males,

as described previously[36]. Males carrying *DsRed* transgenic sperm were subsequently introduced to female cages. After two nights, spermathecae were dissected from females, and the genomic DNA was prepared as described above and subjected to qPCR detection of the *DsRed* transgene; the primers are listed in Extended Data Table 2. The absence of the *DsRed* transgene indicates no occurrence of remating.

### 3D20E synthesis
3D20E was synthesized as previously described[37]. In brief, 10 mg of 20E (Sigma-Aldrich) was dissolved in 10 ml of water, and 30 mg of platinum black (powder form, Sigma-Aldrich) was then added. A gentle stream of $O_2$ gas was bubbled continuously into the reaction mixture, which was stirred at room temperature. After 6 h, the reaction was terminated by adding 30 ml of methanol. The mixture was centrifuged, and the catalyst pellet was removed. The supernatant was evaporated to dryness under vacuum at room temperature. The dried reaction product was dissolved in 10% ethanol for injection and methanol for HPLC–MS/MS analysis. The conversion rate (from 20E to 3D20E) was approximately 97% (Fig. 4b), and the MS spectra of the synthesized 3D20E matched those of 3D20E found in mated females (Fig. 4c).

### Statistical analysis
The figure legends contain specific details of the statistical tests performed. GraphPad (version 9.0) was used to perform Fisher's exact tests, Mantel–Cox tests and Student's *t*-tests. Cochran–Mantel–Haenszel tests were performed using a customized R script (available at https://github.com/duopeng/mantelhaen.test). The normality of data distribution was tested with the Shapiro–Wilk test using a significance threshold of 0.05. Mann–Whitney tests were performed when data failed to pass the normality test. Survival data were analysed with the Mantel–Cox test. The DESeq2 package (version 1.28.1) was used to perform RNA-seq gene-level differential expression analysis. Horizontal bars on graphs represent medians. A significance value of $P = 0.05$ was used as the threshold in all tests.

### Reporting summary
Further information on research design is available in the Nature Research Reporting Summary linked to this paper.

## Data availability
All original western blots are provided in the Supplementary Information.

The MS proteome data were deposited to the ProteomeXchange Consortium (http://proteomecentral.proteomexchange.org) through the PRIDE partner repository (https://www.ebi.ac.uk/pride/) with the dataset identifier PXD032157.

The RNA-seq datasets were deposited to the Gene Expression Omnibus repository (https://www.ncbi.nlm.nih.gov/geo/) under the series record GSE198665.

Other datasets generated during and/or analysed during the current study are available from the corresponding author on reasonable request. Source data are provided with this paper.

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

**Acknowledgements** We thank E. Nelson, K. Thornburg and E. Selland for help with mosquito rearing and the members of the Catteruccia laboratory for comments on the manuscript. Funding for this study was provided by a joint Howard Hughes Medical Institute and Bill and Melinda Gates Foundation (BMGF) Faculty Scholars Award (grant ID OPP1158190) and by the National Institutes of Health (NIH) (award numbers R01 AI104956 and R01 AI124165) to F.C. Proteomics work was supported by NIH (P41 GM103533) to M.J.M. RNA-seq work was supported by the Harvard Data Science Initiative (Postdoctoral Fellow Research Fund, 2019) to D.P. The findings and conclusions within this publication are those of the authors and do not necessarily reflect positions or policies of the HHMI, the BMGF or the NIH. F.C. is an HHMI investigator.

**Author contributions** D.P., E.G.K., S.N.M., W.R.S. and F.C. conceived the study, designed experiments and interpreted data. D.P., E.G.K., S.N.M., K.A. and T.A.R. carried out mating assays. C.V. carried out ecdysteroid mass spectrometry analyses. E.G.K. and S.N.M. carried out RNA-seq sample collection and library preparation. D.P., E.G.K. and S.N.M. analysed the RNA-seq data. D.P. built the customized bioinformatics pipeline. E.M. carried out whole organism isotope labelling and proteome sample collection. G.E.M. and M.J.M carried out proteome mass spectrometry analysis. D.P. performed western blots and coordinated production of the customized antibody. D.P., E.G.K., S.N.M. and T.A.R. carried out dsRNA silencing, gene expression RT–qPCR analysis, oviposition assays and remating assays. D.P. and T.A.R. carried out mortality assays. D.P. performed chemical synthesis, expressed and purified recombinant proteins and performed enzymatic assays. D.P., W.R.S. and F.C. wrote the paper. F.C. supervised the study.

**Competing interests** The authors declare no competing interests.

**Additional information**
**Correspondence and requests for materials** should be addressed to Flaminia Catteruccia.

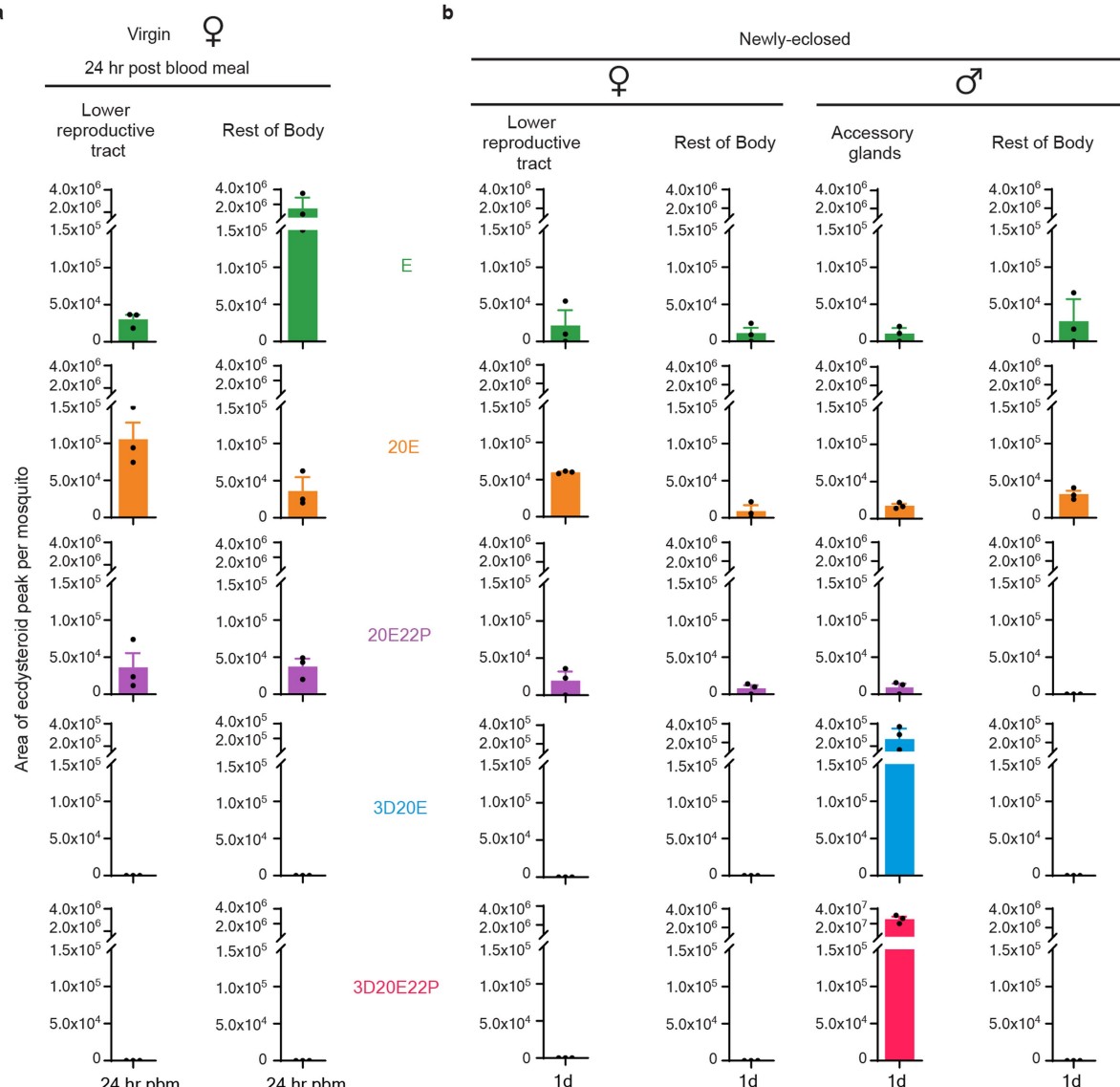

**Extended Data Fig. 1 | (a–b) 3D20E and 3D20E22P are not found in blood-fed and newly-eclosed females while they are specifically found in the MAGs of newly-eclosed males.** 3D20E and 3D20E22P are not detected in (**a**) virgin blood-fed females and (**b, left**) newly-eclosed (within 1 day of eclosion) females but are specifically detected in (**b, right**) the accessory glands (MAGs) of newly-eclosed males. E, 20E and 20E22P are detected at different levels in most samples. **In all panels:** Data are presented as means ± s.e.m. derived from three biological replicates (pbm = post blood meal).

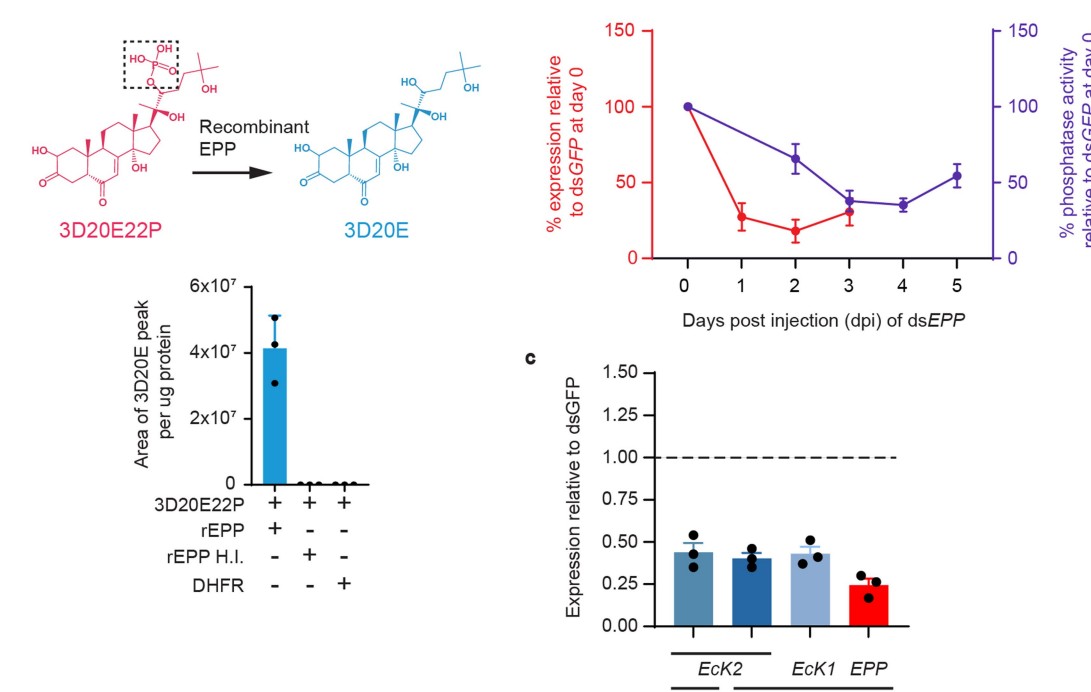

**Extended Data Fig. 2 | Recombinant EPP enzymatic assay and dsRNA silencing efficiencies.** (**a**) Schematic representation of the reaction expected to be catalyzed by sexually transferred EPP in the mated female atrium. Dotted line highlights the phosphate moiety targeted by EPP. Recombinant EPP (rEPP) catalyzes the dephosphorylation of 3D20E22P *in vitro*. Using 3D20E22P (extracted from the MAGs by HPLC-MS/MS) as the substrate, HPLC-MS/MS was used to detect 3D20E production following incubation with the recombinant enzyme. DHFR: dihydrofolate reductase, used as negative control; H.I.: heat-inactivated rEPP, used as a second negative control. Data are presented as

means ± s.e.m. that were derived from three *in vitro* assay replicates. (**b**) Time-course analysis of *EPP* expression (red) and phosphatase activity (purple) in the MAGs relative to ds*GFP* at day 0 (means ± s.e.m.). *EPP* levels were lowest at two days post injection (dpi), and phosphatase activity was lowest at 3–4 dpi. Data were pooled from three replicates. (**c**) RT-qPCR silencing efficiencies for dsRNA injections, normalized to the housekeeping ribosomal gene *RpL19*, and calculated as the expression of the target gene relative to the *GFP* control (means ± s.e.m.). Data were pooled from three replicates.

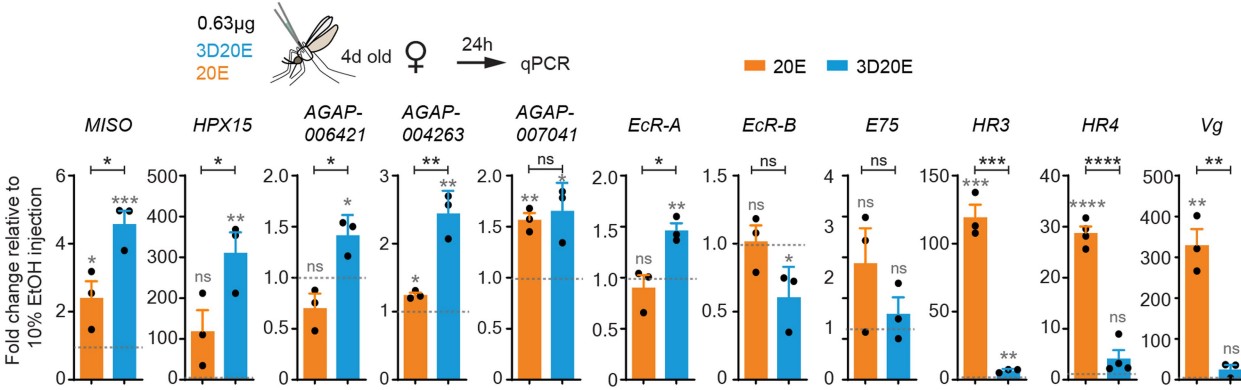

**Extended Data Fig. 3 | Gene expression analysis after ecdysteroid injections.** 3D20E injections (0.63 µg) induce significantly stronger expression of *MISO* compared to the same amount of 20E. 3D20E injections also specifically or more potently activate the expression of *EcR* isoform A (*EcR-A*), and the mating-induced genes *AGAP006421*, *AGAP004263* and *HPX15*, while they repress the *EcR* isoform *EcR-B*. 20E injections specifically activate the yolk protein gene *Vg* and *HR4*, and more potently induce *HR3* expression (Data are presented as means ± s.e.m. that were derived from three biological replicates of pools of 10 mosquitoes each (*HR4* data were from four biological relications of pools of 10 mosquitoes each), unpaired, two-sided, FDR-corrected t-tests were conducted for 20E vs 3D20E comparisons; statistical significance above each bar denotes comparison with 10% EtOH injection: unpaired t-test, two-sided). See Supplementary Table 3 for *P* values.

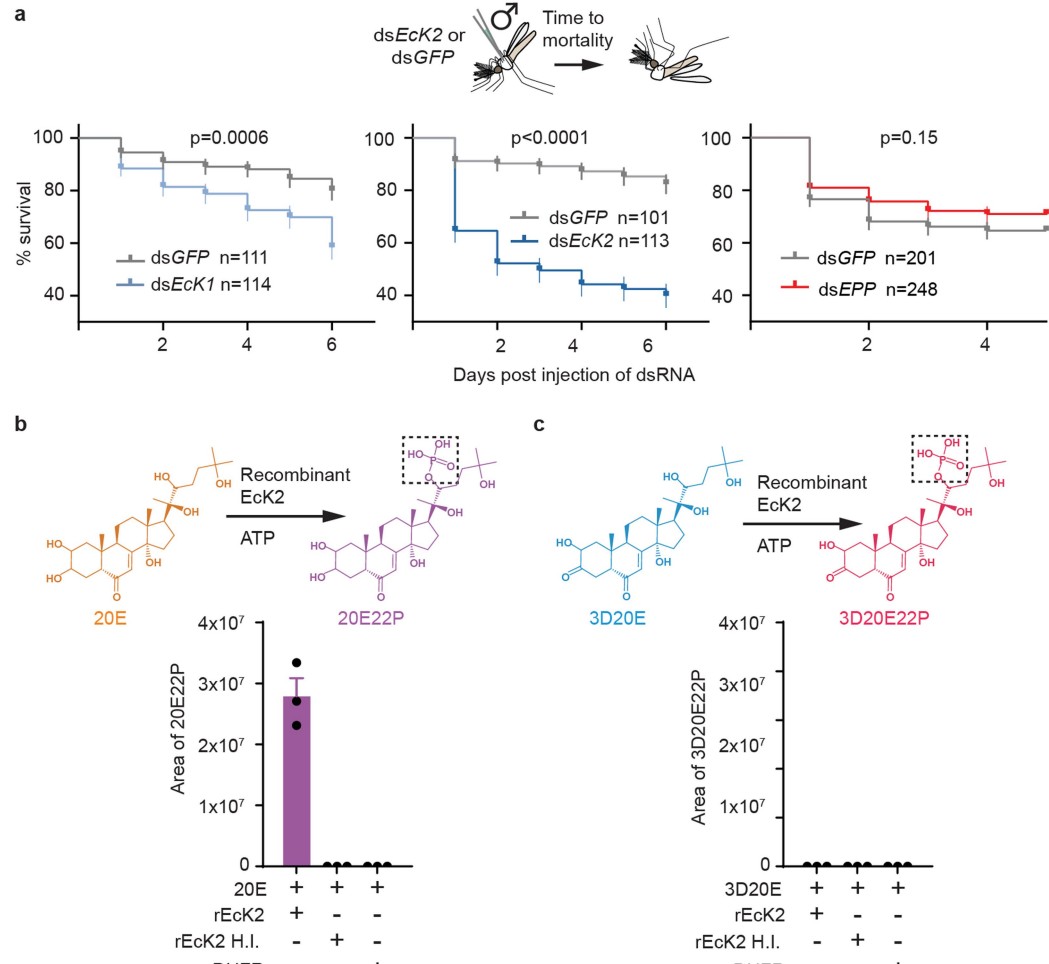

**Extended Data Fig. 4 | Male survival is affected by kinase silencing.**
(**a**) Survival of males is negatively affected by silencing of *EcK1* and *EcK2* but not *EPP*. Survival rate is significantly lowered in *EcK1*-silenced males (left) and *Eck2*-silenced males (middle), but not in *EPP*-silenced (right) males relative to ds*GFP* controls (means ± s.e.m., n indicates the number of biologically independent mosquito samples, Mantel-Cox test, two-sided).

(**b–c**) Recombinant EcK2 (rEcK2) efficiently phosphorylates (**b**) 20E but not (**c**) 3D20E, as determined by HPLC-MS/MS. Means ± s.e.m. were derived from three *in vitro* assay replicates. Negative controls (DHFR and heat-inactivated, H.I., rEcK2) do not phosphorylate 20E or 3D20E. Schematic representations of the reactions catalyzed by EcK2 are provided.

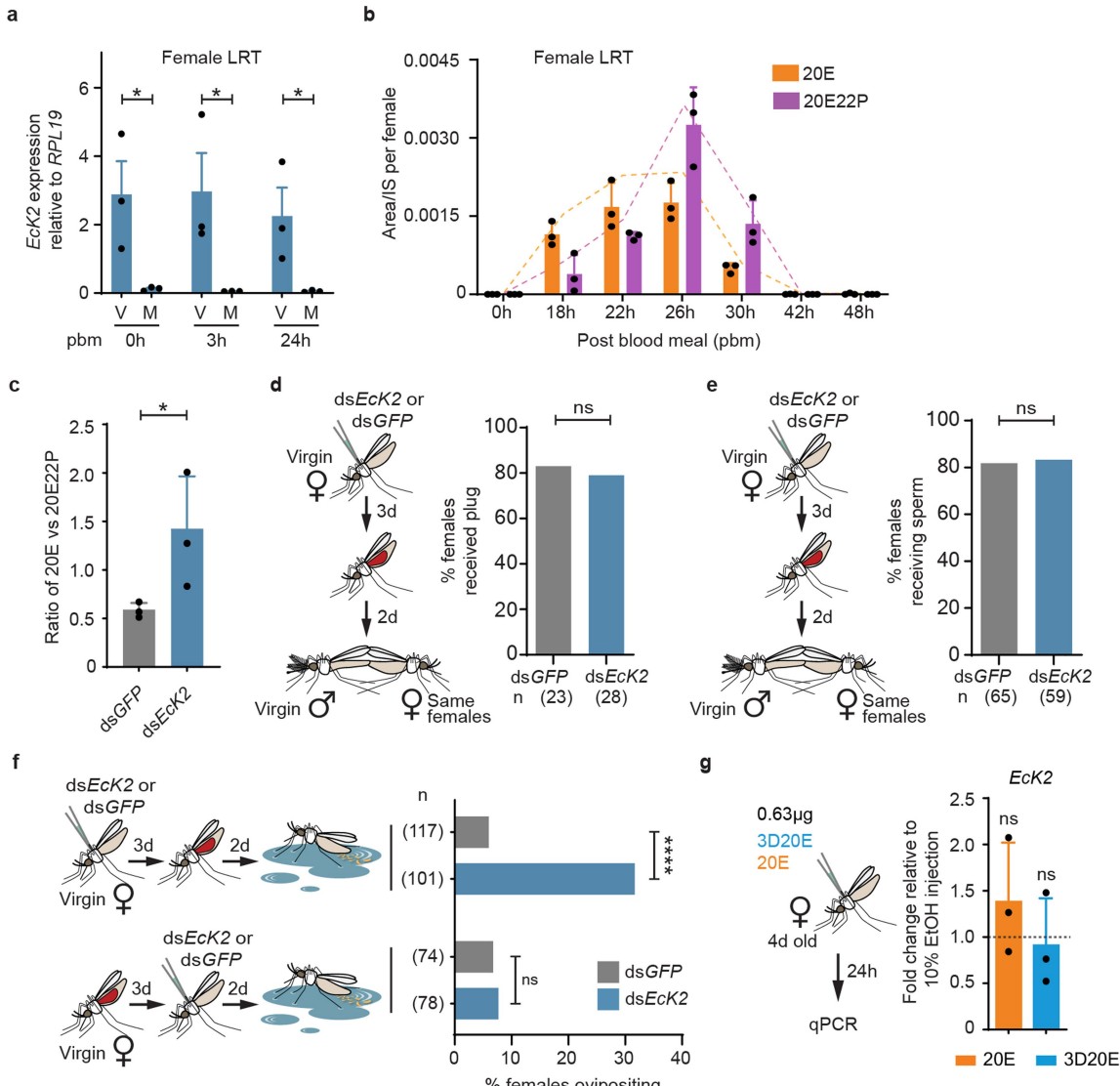

**Extended Data Fig. 5 | EcK2 phosphorylates 20E in blood-fed females and prevents oviposition in virgins.** (**a**) Expression of *EcK2* is high in the lower reproductive tract (LRT) of virgin females and is suppressed after mating. Blood feeding does not affect *EcK2* expression at different time points post blood meal (pbm) (0 h: *P* = 0.024, 3 h: *P* = 0.031, 24 h: *P* = 0.029, means ± s.e.m., unpaired t-test, one-sided). (**b**) 20E22P is detected in the LRT of virgin females after a blood meal (means ± s.e.m.). Quantification was performed using HPLC-MS/MS. Dotted lines connect mean values of the same ecdysteroid at different time points. (**c**) *EcK2* silencing prior to blood feeding significantly increases the ratio of 20E vs 20E22P at 26 pbm in the LRT of virgin females (*P* = 0.047, means ± s.e.m., unpaired t-test, one-sided). (**d**) Mating receptivity (assessed by the presence of a mating plug) in blood-fed virgin females is not affected by *EcK2* silencing (*P* = 0.50, Fisher's exact test, two-sided, data pooled

from two replicates). (**e**) Mating receptivity is not affected in virgins by *EcK2* silencing (*P* = 0.98, Cochran–Mantel–Haenszel test, two-sided). (**f**) Oviposition is highly induced in virgins when *EcK2* is silenced prior to blood feeding (**top**) (*P* = 3.30e-06, Cochran–Mantel–Haenszel test, two-sided, data were from four replicates), but not when this kinase is silenced post blood feeding (**bottom**) (*P* = 0.94, Cochran–Mantel–Haenszel test, two-sided). (**g**) *EcK2* expression is not suppressed by injection of 20E or 3D20E relative to 10% EtOH-injected controls. Expression values were normalized to the housekeeping gene *RpL19* in respective samples before calculating fold changes (means ± s.e.m., unpaired t-tests, two-sided, 20E: *P* = 0.46, 3D20E: *P* = 0.85). **In all panels:** n indicates the number of biologically independent mosquito samples, ns = not significant. * *P* < 0.05, ** *P* < 0.01, *** *P* < 0.001, **** *P* < 0.001. Data were from three biological replicates in all panels unless noted otherwise.

**Extended Data Table 1 | MS/MS quantification of 20E and 3D20E in mosquitoes tissues**

| | Ecdysteroid | Virgin MAG (♂) 4d | Mated LRT (♀) 3 hpm | 24 hr post blood meal (virgin ♀) | | Thorax injection with 630,000 pg ecdysteroid (virgin ♀) | |
| --- | --- | --- | --- | --- | --- | --- | --- |
| | | | | LRT | Rest of Body | LRT 24 hpi | Rest of Body 24 hpi |
| pg detected per animal | 20E | 86 | 18 | 31 | 12 | 361 (20E injection) | 24,694 (20E injection) |
| | 3D20E | 211 | 2202 | 0 | 0 | 1066 (3D20E injection) | 35,363 (3D20E injection) |

(hpm = hours post mating, hpi = hours post injection).

**Extended Data Table 2 | Primer sequences used in qPCR, RT–qPCR and the cloning of plasmid constructs for generation of dsRNA and overexpression of recombinant proteins.**

| | Forward primer | Reverse primer | Product Size |
|---|---|---|---|
| EPP (RT-qPCR) | CACGTTTCCTTCCGTGCTTT | TACAAACCGCTCTCTTCGCC | 120bp |
| EcK1 (RT-qPCR) | TCGAACCAGGCGTTAGTCAA | CCAGAAGATCTGGAAGAATACTGGT | 141bp |
| EcK2 (RT-qPCR) | CTACTACTCGGCCTCAACC | CTGCTGCTCAATCACTCAC | 79bp |
| MISO (RT-qPCR) | AGACGATGGAGGGACTGATG | GGATTCGCTTTCGTGCTG | 80bp |
| HPX15 (RT-qPCR) | GTTCGCAACGGCTGCATTCC | TGTTGAACCAATCGGACAGACGG | 113bp |
| AGAP006421 (RT-qPCR) | TTCTTCGGCTACCGGTTCAC | TCCGATCGGTTTTCCACTGT | 132bp |
| AGAP007041 (RT-qPCR) | TTCTGCAGTGTTACGATGCG | TGCAGCGACTCGAATGAACC | 109bp |
| AGAP004263 (RT-qPCR) | ACATGACAATGGTGAACTGCG | CGAGTGGTATCGAACGTAGCC | 118bp |
| EcR-A (RT-qPCR) | AACGGTGGCAACAATCTGAAC | CAGATTGCCCACGAACTGATG | 120bp |
| EcR-B (RT-qPCR) | ACACGATGTCGAACGGCTAC | TCGACGGGGACAAATCTTCA | 99bp |
| E75 (RT-qPCR) | TGCGAAGGATGCAAGGGTTT | GCACTGCTGATTTTTCGTGC | 81bp |
| HR3 (RT-qPCR) | GGAATGAGTCGTGACGCTGT | GTGAAACCGTACCTCGTCCT | 81bp |
| HR4 (RT-qPCR) | CTGCAACACACTGTTCCACA | GCTGCCGTTTGAATCTCTGG | 77bp |
| Vg (RT-qPCR) | CCGACTACGACCAGGACTTC | CTTCCGGCGTAGTAGACGAA | 118bp |
| Broad (RT-qPCR) | CGTGAAGGGCTCTTTGCGAT | ATGAGAAACTGTGCCCCGC | 81bp |
| E74 (RT-qPCR) | GAGGTAGCGAACCTGTGCTG | CACGTTACAGCCGTCCCAG | 119bp |
| RPL19 (RT-qPCR) | CCAACTCGCGACAAAACATTC | ACCGGCTTCTTGATGATCAGA | 61bp |
| EPP (dsRNA region) | GCAACAGAACGGTGGCAAAAT | GGACGAGCCGTGCCTGATAG | 362bp |
| EcK1 (dsRNA region) | TTTCTCTTTCAAACGACACGG | GTGTGCTGACGTCGAGTGTT | 365bp |
| EcK2 (dsRNA region) | TAGCGGTGAGTGATTGAGCA | ACCGATTTCGCCTTACAGTG | 556bp |
| T7 promoter | TAATACGACTCACTATAGGG | N.A. | N.A. |
| EPP (overexpression) | ATGTTCGGTGGCAATAAAACGATGGACGACATTCCACT | CTCTAATAGTATCTTGTAGTCAAACCGGTTGTTGTTGGTGTGCGTCA | 960bp |
| EcK2 (overexpression) | ATGCAAGCCCGCACAGCCGTCAGGATGAAGCAAGAAGTGGCGG | GATGCGGTCGCTCTTGAGCCGCTGCAGCACGCCTTGATCGTA | 1380bp |
| dsRed (genomic qPCR) | ATGGTGCGCTCCTCCAAGAACG | ACCTTCAGCTTCACGGTGTTGTGG | 146bp |
| Y (genomic qPCR) | GGATCTGGCCAAGAGGAGTA | CCCAACCAAGGTACTCTAACG | 148bp |

# Reporting Summary

## Statistics

For all statistical analyses, confirm that the following items are present in the figure legend, table legend, main text, or Methods section.

| n/a | Confirmed | |
|-----|-----------|---|
| ☐ | ☒ | The exact sample size (*n*) for each experimental group/condition, given as a discrete number and unit of measurement |
| ☐ | ☒ | A statement on whether measurements were taken from distinct samples or whether the same sample was measured repeatedly |
| ☐ | ☒ | The statistical test(s) used AND whether they are one- or two-sided *Only common tests should be described solely by name; describe more complex techniques in the Methods section.* |
| ☒ | ☐ | A description of all covariates tested |
| ☐ | ☒ | A description of any assumptions or corrections, such as tests of normality and adjustment for multiple comparisons |
| ☐ | ☒ | A full description of the statistical parameters including central tendency (e.g. means) or other basic estimates (e.g. regression coefficient) AND variation (e.g. standard deviation) or associated estimates of uncertainty (e.g. confidence intervals) |
| ☐ | ☒ | For null hypothesis testing, the test statistic (e.g. *F*, *t*, *r*) with confidence intervals, effect sizes, degrees of freedom and *P* value noted *Give P values as exact values whenever suitable.* |
| ☒ | ☐ | For Bayesian analysis, information on the choice of priors and Markov chain Monte Carlo settings |
| ☒ | ☐ | For hierarchical and complex designs, identification of the appropriate level for tests and full reporting of outcomes |
| ☒ | ☐ | Estimates of effect sizes (e.g. Cohen's *d*, Pearson's *r*), indicating how they were calculated |

*Our web collection on statistics for biologists contains articles on many of the points above.*

## Software and code

Policy information about availability of computer code

| | |
|---|---|
| Data collection | Xcalibur 4.4, Design and Analysis 2.4.3, Image Studio 5.2 |
| Data analysis | Xcalibur 4.4, Tracefinder 4.1, Proteowizard 3.0.20287, Comet 3.2, Percolator 3.05, Limelight 2.2.0, Image Studio 5.2, Design and Analysis 2.4.3, Graphpad 9.0., HISAT2 2.0.5, Samtools 1.3.1, htseq-count 0.9.1, DESeq2 1.28.1, R 4.0.3, PSI-Blast 2.8.1, Primer-BLAST (https://www.ncbi.nlm.nih.gov/tools/primer-blast/) |

For manuscripts utilizing custom algorithms or software that are central to the research but not yet described in published literature, software must be made available to editors and reviewers. We strongly encourage code deposition in a community repository (e.g. GitHub). See the Nature Portfolio guidelines for submitting code & software for further information.

## Data

Policy information about availability of data

All manuscripts must include a data availability statement. This statement should provide the following information, where applicable:
- Accession codes, unique identifiers, or web links for publicly available datasets
- A description of any restrictions on data availability
- For clinical datasets or third party data, please ensure that the statement adheres to our policy

All original western blots are provided in the Supplementary Information.
The MS proteome data were deposited to the ProteomeXchange Consortium (http://proteomecentral.proteomexchange.org) via the PRIDE partner repository (https://www.ebi.ac.uk/pride/) with the dataset identifier PXD032157.
The RNAseq data were deposited to the Gene Expression Omnibus repository (https://www.ncbi.nlm.nih.gov/geo/) under the series record GSE198665.
Other datasets generated during and/or analysed during the current study are available from the corresponding author on reasonable request.

# Field-specific reporting

Please select the one below that is the best fit for your research. If you are not sure, read the appropriate sections before making your selection.

☒ Life sciences          ☐ Behavioural & social sciences          ☐ Ecological, evolutionary & environmental sciences

For a reference copy of the document with all sections, see nature.com/documents/nr-reporting-summary-flat.pdf

# Life sciences study design

All studies must disclose on these points even when the disclosure is negative.

| | |
|---|---|
| Sample size | We did not perform power analysis for sample size calculation. Sample size was shaped by a combination of time, feasibility, and prior experience to obtain maximum statistical power with reasonable resource. The following factors had a large impact on sample size: (1) the maximum number of mosquitoes housed in a laboratory cage (small cage n=100, medium cage n=200), (2) blood-feeding / mating rates, injection survival rates, and (3) the maximum number of mosquito can be processed/dissected for each time point. |
| Data exclusions | No data were excluded from the analysis |
| Replication | The experimental findings were confirmed with three or more replicates with the exception of western blot (2 biological replicates). |
| Randomization | Mosquitoes were randomly allocated into experimental groups by aspiration |
| Blinding | Investigators were blinded to group allocation during data analysis. The investigators were not blinded to group allocation during mosquito sample processing, and the mosquito cages were labeled with group allocation information that can be seen by the investigators. |

# Reporting for specific materials, systems and methods

We require information from authors about some types of materials, experimental systems and methods used in many studies. Here, indicate whether each material, system or method listed is relevant to your study. If you are not sure if a list item applies to your research, read the appropriate section before selecting a response.

## Materials & experimental systems

| n/a | Involved in the study |
|---|---|
| ☐ | ☒ Antibodies |
| ☒ | ☐ Eukaryotic cell lines |
| ☒ | ☐ Palaeontology and archaeology |
| ☐ | ☒ Animals and other organisms |
| ☒ | ☐ Human research participants |
| ☒ | ☐ Clinical data |
| ☒ | ☐ Dual use research of concern |

## Methods

| n/a | Involved in the study |
|---|---|
| ☒ | ☐ ChIP-seq |
| ☒ | ☐ Flow cytometry |
| ☒ | ☐ MRI-based neuroimaging |

## Antibodies

| | |
|---|---|
| Antibodies used | Primary Antibody:<br>(1) Anti-EPP, custom ordered from Genscript NJ, US (see methods for details)<br>(2) Anti-Actin antibody [MAC 237] (ab50591) Abcam<br><br>Secondary Antibody:<br>(1) IRDye® 800CW Donkey anti-Rabbit IgG Secondary Antibody, P/N: 926-32213<br>(2) IRDye® 680LT Goat anti-Rat IgG Secondary Antibody, P/N: 926-68029 |
| Validation | Anti-EPP was validated by western blot using recombinant EPP and male accessory glands samples. Cross-reactivity was verified by western using virgin female mosquito samples.<br>Anti-Actin antibody was validated for use in Anopheles gambiae mosquitoes by Werling et al., 2019 Cell 177, 315-325.<br>Secondary antibodies were confirmed by Werling et al., 2019 Cell 177, 315-325. |

## Animals and other organisms

Policy information about studies involving animals; ARRIVE guidelines recommended for reporting animal research

| | |
|---|---|
| Laboratory animals | Anopheles gambiae G3 strain, male and female, age: 1 day old and 4 days old. |
| Wild animals | No wild animals were used in this study |

| Field-collected samples | No field-collected samples were used in this study |
| Ethics oversight | No ethical approval or guidance was required |

Note that full information on the approval of the study protocol must also be provided in the manuscript.

