## [Peer Review File · Nature]

Manuscript Title: A male steroid controls female sexual behavior in the malaria mosquito

Reviewer Comments & Author Rebuttals

Reviewer Reports on the Initial Version:

Referees' comments:

Referee #1 (Remarks to the Author):

The manuscript by Peng et al. describes the identification of the first insect sex-specific steroid hormone and its behavioral function in the human malaria vector mosquito *Anopheles gambiae*. It has been known for some time, that this insect becomes refractory to secondary rounds of mating after an initial copulation transfers sperm and other components produced by the male accessory gland to receptive females, but the molecular mechanisms behind the behavioral change has remained elusive. In the present study, the authors demonstrate that male *A. gambiae* produce 3D20E22P, an oxidized and phosphorylated form of 20E, in the male accessory gland and that this hormone is transferred to the female lower reproductive tract (LRT) during mating along with an ecdysteroid phosphatase (EPP). After transfer, dephosphorylation of 3D20E22P in the LRT by the transferred EPP lead to accumulation of 3D20E and induction of mating refractory behavior. That 3D20E is the primary active compound was demonstrated via direct injection of 3D20E into virgin females which then exhibit refractory mating behavior. The authors provide several convincing controls which demonstrate that the precursors E and 20E do not elicit this behavior. The authors also identify the Ecdysone Kinase likely responsible for phosphorylation of 3D20E to 3D20E22P but were unable to find the Ecdysone Oxidase responsible for oxidizing 20E to 3D20E. Nevertheless, this is a compelling story with significant implications for potential development of novel strategies for manipulating reproductive success of malaria-transmitting mosquitoes.

Overall, I found the study to be thorough and convincing with appropriate methods and statistics. Of course, there are several key unanswered questions such as how the differential effects of 3D20E in terms of transcriptional induction profiles are elicited compared to 20E (Fig 4F). Does this hormone have different agonistic activities using standard 20E transcriptional response assays, for example, or might it involve a non-genomic response using a putative 20E responsive GPCR. Another important question is the identity and expression/activity profile of the Ecdysone Oxidase necessary to produce 3D20E. I think both issues can be left for future studies, since they don't detract much from the key findings, but I am curious if the authors actually examined by in situ the expression of the three putative *A. gambiae* EO homologs, just in case their expression profiling pipeline missed something.

Michael B O'Connor

Referee #2 (Remarks to the Author):

This paper reports the identification of a modified ecdysteroid (3D20E) that male *Anopheles gambiae* transfer to females on mating. Receipt of this substance inhibits female receptivity to mating, and full activation of this effect requires a male derived phosphatase. Its effects seem to be mainly on receptivity rather than oviposition per se.

How novel is this result? It is certainly an advance, as the male specific ecdysteroid 3D20E was previously unknown, and it is clearly much more potent than 20E. The finding that males also transfer a modifying enzyme that seems to activate it is very neat too. But previous work from the group on (non sex specific) 20E made similar claims previously: "Sexual transfer of the steroid hormone 20E induces the postmating switch in *Anopheles gambiae*" PNAS 2014. I would suggest that the current work IS a big enough advance to warrant publication in Nature (and the previous work was perhaps somewhat oversold). It really does seem to have identified a substance something akin to the potency of sex peptide in *Drosophila*. Related to that, it seems pretty odd that the parallel *Drosophila* system isn't discussed in the ms – and the work isn't really placed in the wider context of research on insect seminal molecules. MAG substances have been known to cause refractoriness and induce oviposition in female insects for many decades (with the first specific molecule, *Drosophila* sex peptide, being discovered in the 80s). And male and female derived modifiers to male seminal proteins have been shown previously in *Drosophila*, so conceptually that is not a new thing per se. It'd perhaps be better to hear about how the work fits into that history, rather than about how deadly malaria is (we all know malaria is a very bad thing).

The data is convincing, and well presented. I see no issues with the methods or stats, and I don't see any reason to worry that the data would prove hard to repeat.

Beyond the lack of context of the work in the wider field of insect reproductive molecules, there are a few oddities in the text.

L23. The opening sentence is quite odd. Firstly 3D20E doesn't really function in males – males manufacture it, but its function is in females after transfer. I'm not sure that identification of sex specific form of an ecdysteroid is that big a selling point (although maybe others would disagree).

L27 "As egg development and mating are essential for species survival...". This is a pretty facile sentence – you could replace 'egg development' or 'mating' with any trait (legs, brain, eating, flying etc...), and that statement would still be true.

L34 (and elsewhere). "...and ensures fertility". It'd be more accurate to say that 3D20E ensures paternity. 3D20E prevents sperm competition from rivals. Without it females might happily mate with multiple males until they are fully fertile, or maybe even more fertile than normal (e.g. see "Monogamy and the battle of the sexes", Hosken et al 2009).

Several references are made to the potential for this work to lead to better mosquito/malaria control strategies. This maybe true but probably now through the mechanisms suggested. I don't see much potential for reducing the reproductive success of mosquitos with the newly discovered molecules. Selection would quickly find ways around any attempts to make less fertile males or females. Perhaps the the main value I would envisage would be in using the knowledge to ensure

that released males are competitive or extra-competitive, post-mating, in any SIT schemes with this species (the potential for SIT in *Anopheles* has certainly been suggested by others). That would still be a very important potential application.

Referee #3 (Remarks to the Author):

This manuscript reports two new and important findings for the reproductive biology of *Anopheles gambiae* the most important African malaria vector. Understanding its biology can lead to new targets for vector control, reducing the burden of malaria. The first finding is that a component of the male accessory gland (AG) secretions (3D20E) renders females refractory to mating and increases fecundity. The second important finding is that mating relieves blocks on oviposition (by a kinase Eck2 which phosphorylates 20E and potentially other factors) in order for female-derived 20E to induce oviposition after blood ingestion. The pathways leading to 3D20E synthesis and activation in this study could be targeted for new control strategies.

The study is exceptionally well designed and conducted with multiple confirmatory assessments of the findings using different tools along the way. It represents a massive amount of work. The authors employed HPLC-MS-MS to detect male-specific steroids in the male AG and the female atrium after transfer. Performed proteomics with isotopic labelling to identify MAG proteins and peptides. They purified 3D20E22P from the MAGs, confirmed it as the most abundant male ecdysteroid transferred to females, identified modifying kinase candidates by transcriptome screening and dsRNA silencing, performed RNAseq to identify function and found that EPP also produced and transferred during mating dephosphorylates 3D20E22P to an active form to induce mating-refractory behavior in females. Overall, this work clarifies the authors' prior work regarding the role of 20E and provides further details on ecdysteroids and their interplay between the sexes in an important malaria vector.

The work is novel and convincing with a high level of significance to those working in the field of vector biology as well as those with a broader interest in reproductive biology. Relevant sources are cited. The data presented are replicated and raw data for presented and extended data are available as google sheets by the authors.

Minor comments and suggestions are as follows:

Line 67- This claim is surprising and may need to be nuanced or re-written for clarity. Part of the confusion is the intersection between male and female reproductive functions which rely on each other.

Line 82 - "We also profiled ecdysteroids in newly eclosed males and females and detected 3D20E and 3D20E22P only in the MAGs, while E, 20E and 20E22P were present in both sexes "
Hyphenate newly-eclosed females both in text and figure legend. The precise timing of this

measurement is not detailed in the text, methods, or figure legend. Is this in the first few hours, first day? Also, the X-axis label for Extended Figure 1B reads "4d" for the "newly-eclosed females".

Lines 96-101 - For the bioinformatics pipeline to deduce possible 20E-modifying enzymes (Fig. 2A), both the second step and the final step reference reproductive tissue specificity as a criteria. Please clarify the differences for these steps.

Line 123 - "Oviposition of eggs (another response triggered in females by mating) was instead unaffected".

This figure shows percentage of females that oviposit, but are there any differences in the number of eggs laid between dsEPP RNAi knockdown and dsGFP knockdown? These data aren't shown.

Line 132-133 - "3D20E was significantly more effective than 20E in turning off the female's susceptibility to mating at two concentrations".

Any comment about how the injected amounts compare to what was determined via MS/MS?

Line 150-151 - "Interestingly, 3D20E triggered other mating-induced genes including HPX15, which affects female fertility".

This refers to Fig. 4f (which is mislabeled in the text as 3f). HPX15 was previously shown to be upregulated significantly via 20E in previous work, yet here the 20E experiment (orange) it is noted as not significant (ns). The y-axis for this is well over 100-fold change over vehicle injection alone, so we presume this is an error.

Line 172 - "Silencing of Eck2 in virgin females caused a 3-fold increase in the relative ratio of 20E to 20E22P after blood feeding".

Level of knockdown was presented for dsEPP, but no data or comments on the level of knockdown for Eck2 is presented. It needs to be included to support the statement on Line 172. Also, here in text the time scale after blood feeding is not given, though described as 26 h in the figure legend. This time should be added in text as well, particularly since this time point in a WT female corresponds to less 20E and more 20E22P.

Fig 5 and line 352. Data were pooled in all panels. Was this statistically appropriate to combine the data? For example, how were pooled % displayed if there was variation between replicates?

Author Rebuttals to Initial Comments:

Referees' comments:

Referee #1 (Remarks to the Author):

The manuscript by Peng et al. describes the identification of the first insect sex-specific steroid hormone and its behavioral function in the human malaria vector mosquito *Anopheles gambiae*. It has been known for some time, that this insect becomes refractory to secondary rounds of mating after an initial copulation transfers sperm and other components produced by the male accessory gland to receptive females, but the molecular mechanisms behind the behavioral change has remained elusive. In the present study, the authors demonstrate that male *A. gambiae* produce 3D20E22P, an oxidized and phosphorylated form of 20E, in the male accessory gland and that this hormone is transferred to the female lower reproductive tract (LRT) during mating along with an ecdysteroid phosphatase (EPP). After transfer, dephosphorylation of 3D20E22P in the LRT by the transferred EPP lead to accumulation of 3D20E and induction of mating refractory behavior. That 3D20E is the primary active

compound was demonstrated via direct injection of 3D20E into virgin females which then exhibit refractory mating behavior. The authors provide several convincing controls which demonstrate that the precursors E and 20E do not elicit this behavior. The authors also identify the Ecdysone Kinase likely responsible for phosphorylation of 3D20E to 3D20E22P but were unable to find the Ecdysone Oxidase responsible for oxidizing 20E to 3D20E. Nevertheless, this is a compelling story with significant implications for potential development of novel strategies for manipulating reproductive success of malaria-transmitting mosquitoes.

Overall, I found the study to be thorough and convincing with appropriate methods and statistics.

We are delighted that Dr. O'Connor made such positive comments on our study.

Of course, there are several key unanswered questions such as how the differential effects of 3D20E in terms of transcriptional induction profiles are elicited compared to 20E (Fig 4F). Does this hormone have different agonistic activities using standard 20E transcriptional response assays, for example, or might it involve a non-genomic response using a putative 20E responsive GPCR.

This is an excellent question that we have been thinking about for a while now. We have some preliminary evidence that 3D20E does not act via the standard nuclear EcR/USP receptor, also based on the transcriptional data from Fig. 4. We suspect, that, similar to *Drosophila* Sex- peptide, 3D20E may trigger a non-genomic response upon interacting with a GPCR on the female reproductive tract. In future studies it would be very interesting to identify the 3D20E receptor and determine whether the two systems (based on steroid hormones vs sex peptides) share downstream signaling cascades.

Another important question is the identity and expression/activity profile of the Ecdysone Oxidase necessary to produce 3D20E. I think both issues can be left for future studies, since they don't detract much from the key findings, but I am curious if the authors actually examined by *in situ* the expression of the three putative *A. gambiae* EO homologs, just in case their expression profiling pipeline missed something.

We did not examine the *in situ* expression of the three putative *An. gambiae* EO homologs. However, despite the fact that they did not emerge from our bioinformatic pipeline, we silenced (by dsRNA injection) these genes in males and checked by MS/MS if there were any effects on 20E oxidation. As we did not find any changes in 20E oxidation levels in single or triple co-silencing of the three EO homologs, we did not proceed with these analyses. We are very interested in finding the oxidizing enzyme/s, but we suspect this will take some time given the best candidates do not appear to be involved in this process. Due to space constraints and also based on Dr. O'Connor's comments above, we will leave these data out of the manuscript as they do not add to the main story.

Michael B O'Connor

Thank you for signing your response.

Referee #2 (Remarks to the Author):

This paper reports the identification of a modified ecdysteroid (3D20E) that male *Anopheles gambiae* transfer to females on mating. Receipt of this substance inhibits female receptivity to mating, and full activation of this effect requires a male derived phosphatase. Its effects seem to be mainly on receptivity rather than oviposition per se.

How novel is this result? It is certainly an advance, as the male specific ecdysteroid 3D20E was previously unknown, and it is clearly much more potent than 20E. The finding that males also transfer a modifying enzyme that seems to activate it is very neat too. But previous work from the group on (non sex specific) 20E made similar claims previously: "Sexual transfer of the steroid hormone 20E induces the postmating switch in *Anopheles gambiae*" PNAS 2014. I would suggest that the current work IS a big enough advance to warrant publication in Nature (and the previous work was perhaps somewhat oversold). It really does seem to have identified a substance something akin to the potency of sex peptide in *Drosophila*.

We are very pleased that the reviewer found our study significantly advances the field.

Related to that, it seems pretty odd that the parallel *Drosophila* system isn't discussed in the ms – and the work isn't really placed in the wider context of research on insect seminal molecules. MAG substances have been known to cause refractoriness and induce

oviposition in female insects for many decades (with the first specific molecule, *Drosophila* sex peptide, being discovered in the 80s). And male and female derived modifiers to male seminal proteins have been shown previously in *Drosophila*, so conceptually that is not a new thing per se. It'd perhaps be better to hear about how the work fits into that history, rather than about how deadly malaria is (we all know malaria is a very bad thing).

The reviewer raises a very good point. Initially we introduced a comparison with the *Drosophila* seminal fluid system, but we took it off given the space constraints of *Nature*. We have now added a short discussion of the similarities and differences between the two systems, to place this work into the bigger picture of post-mating responses in insects.

The data is convincing, and well presented. I see no issues with the methods or stats, and I don't see any reason to worry that the data would prove hard to repeat.

Beyond the lack of context of the work in the wider field of insect reproductive molecules, there are a few oddities in the text.

L23. The opening sentence is quite odd. Firstly 3D20E doesn't really function in males – males manufacture it, but its function is in females after transfer. I'm not sure that identification of sex specific form of an ecdysteroid is that big a selling point (although maybe others would disagree).

We agree the 3D20E functions we identified are in the female, so we have changed this sentence to a more generic one: *Insects, unlike vertebrates, are widely believed to lack male-biased sex steroid hormones*. However we'd like to point out that although its function is in the female, 3D20E is a male steroid as females do not produce it, and its functions benefit males by enforcing monogamy. This is the first identification of a male steroid, changing the paradigm that insects do not have them.

L27 "As egg development and mating are essential for species survival...". This is a pretty facile sentence – you could replace 'egg development' or 'mating' with any trait (legs, brain, eating, flying etc...), and that statement would still be true.

We agree with the reviewer and have changed this sentence to: *as egg development and mating are essential reproductive traits...*

L34 (and elsewhere). "...and ensures fertility". It'd be more accurate to say that 3D20E ensures paternity. 3D20E prevents sperm competition from rivals. Without it females might happily mate with multiple males until they are fully fertile, or maybe even more fertile than normal (e.g. see "Monogamy and the battle of the sexes", Hosken et al 2009).

Our reference to fertility was based on the data showing that reduced 3D20E levels are associated with reduced fertility in the female (Figure. 3f). However, as the fertility data are not at the center of this work, and considering we refer to paternity in the previous sentence, we have removed these few words to avoid confusion.

Several references are made to the potential for this work to lead to better mosquito/malaria control strategies. This maybe true but probably now through the mechanisms suggested. I don't see much potential for reducing the reproductive success of mosquitos with the newly discovered molecules. Selection would quickly find ways around any attempts to make less fertile males or females. Perhaps the the main value I would envisage would be in using the knowledge to ensure that released males are competitive or extra-competitive, post-mating, in any SIT schemes with this species (the potential for SIT in Anopheles has certainly been suggested by others). That would still be a very important potential application.

We agree that knowledge derived from this study provides opportunities for SIT-based releases and have now made a specific reference to this control strategy. However, we believe that also mimicking the function of 3D20E in virgin females could lead to mosquito sterilization, and have therefore left this part in the discussion. It is true that at some point selective pressures would stimulate the emergence of resistance mechanisms. This is true also for insecticide-based control strategies that aim to reduce the viability of insects, but nevertheless these strategies are still the most effective at curbing disease transmission.

Referee #3 (Remarks to the Author):

This manuscript reports two new and important findings for the reproductive biology of *Anopheles gambiae* the most important African malaria vector. Understanding its biology can lead to new targets for vector control, reducing the burden of malaria. The first finding is that a component of the male accessory gland (AG) secretions (3D20E) renders females refractory to mating and increases fecundity. The second important finding is that mating relieves blocks on oviposition (by a kinase Eck2 which phosphorylates 20E and potentially other factors) in order for female-derived 20E to induce oviposition after blood ingestion. The pathways leading to 3D20E synthesis and activation in this study could be targeted for new control strategies.

The study is exceptionally well designed and conducted with multiple confirmatory assessments of the findings using different tools along the way. It represents a massive amount of work. The authors employed HPLC-MS-MS to detect male-specific steroids in the male AG and the female atrium after transfer. Performed proteomics with isotopic labelling to identify MAG proteins and peptides. They purified 3D20E22P from the MAGs, confirmed it as the most abundant male ecdysteroid transferred to females, identified modifying kinase candidates by transcriptome screening and dsRNA silencing, performed RNAseq to identify function and found that EPP also produced and transferred during

mating dephosphorylates 3D20E22P to an active form to induce mating-refractory behavior in females. Overall, this work clarifies the authors' prior work regarding the role of 20E and provides further details on ecdysteroids and their interplay between the sexes in an important malaria vector.

The work is novel and convincing with a high level of significance to those working in the field of vector biology as well as those with a broader interest in reproductive biology. Relevant sources are cited. The data presented are replicated and raw data for presented and extended data are available as google sheets by the authors.

We thank the reviewer for these very positive comments.

Minor comments and suggestions are as follows:

Line 67- This claim is surprising and may need to be nuanced or re-written for clarity. Part of the confusion is the intersection between male and female reproductive functions which rely on each other.

A similar point was also raised by reviewer 1. For clarity, we have rewritten the sentence as follows: *However, while vertebrates have multiple classes of largely dimorphic steroid hormones like estrogens and androgens (reviewed in ⁷), male-biased steroids have never been identified in insects.*

Line 82 - "We also profiled ecdysteroids in newly eclosed males and females and detected 3D20E and 3D20E22P only in the MAGs, while E, 20E and 20E22P were present in both sexes ". Hyphenate newly-eclosed females both in text and figure legend. The precise timing of this measurement is not detailed in the text, methods, or figure legend. Is this in the first few hours, first day? Also, the X-axis label for Extended Figure 1B reads "4d" for the "newly-eclosed females".

Thank you for pointing out these issues. The newly-eclosed mosquitoes were sampled within the first day. We have added this information (and the hyphenation) in the legend. We also corrected the x-axis label.

Lines 96-101 - For the bioinformatics pipeline to deduce possible 20E-modifying enzymes (Fig. 2A), both the second step and the final step reference reproductive tissue specificity as a criteria. Please clarify the differences for these steps.

We realize we did not discuss the pipeline exhaustively. Briefly, the second step requires genes to be expressed at a minimum of the 85th percentile in *An. gambiae* reproductive tissues. In the final step instead, genes must demonstrate additional reproductive specificity by satisfying at least one of the following: (1) they must be significantly upregulated post-mating (in the reproductive tissues), and/or (2) must not be expressed at high levels (over the 85th percentile) in non-reproductive tissues. We added a bioinformatic pipeline section in the methods to provide more information.

Line 123 - "Oviposition of eggs (another response triggered in females by mating) was instead unaffected". This figure shows percentage of females that oviposit, but are there any differences in the number of eggs laid between dsEPP RNAi knockdown and dsGFP knockdown? These data aren't shown.

This is a very good question. We checked the data and found a small but statistically significant decrease in the number of eggs laid in the dsEPP RNAi group. We have now added these data into Figure 3.

Line 132-133 - "3D20E was significantly more effective than 20E in turning off the female's susceptibility to mating at two concentrations". Any comment about how the injected amounts compare to what was determined via MS/MS?

These calculations were originally provided in Extended Data Table 1 but we realize we did not emphasize the data enough. In the LRT, after injection we detected 1066 pg of 3D20E compared to 2202 pg after mating, so half as abundant. For 20E, we detected 361 pg after injections, therefore 20 times more abundant than the amount (18 pg) detected after mating (and 10 times the amount of 20E detected after blood feeding). These data are difficult to interpret as there are still many unknowns, including where the 3D20E receptor is and how quickly the two hormones are metabolized, so we wouldn't want to speculate too much on their significance. However they may suggest that injected 20E gets metabolized more quickly than 3D20E in the LRT. Future work will be needed to understand the full complexity of the system, but for now we have added a paragraph to broadly discuss the data: *Notably, 24 h after injections at the highest concentration, half the physiological level of 3D20E in the LRT (1066 pg after injections vs 2022 pg after mating) induced a greater proportion of refractory females than 20 times the physiological level of 20E (361 pg after injections vs 18 pg after mating) (Extended Data Table 1), in agreement with the notion that sexual transfer of 20E does not induce mating refractoriness and further pointing to 3D20E as the principal factor ensuring paternity.*

Line 150-151 - "Interestingly, 3D20E triggered other mating-induced genes including HPX15, which affects female fertility". This refers to Fig. 4f (which is mislabeled in the text as 3f). HPX15 was previously shown to be upregulated significantly via 20E in previous work, yet here the 20E experiment (orange) it is noted as not significant (ns). The y-axis for this is well over 100-fold change over vehicle injection alone, so we presume this is an error.

We corrected the figure reference mistake in the main text. Our statistics are however correct: *HPX15* upregulation is not significant after 20E injection. Although the effect size is considerable (100-fold), the high variance (large error bar) prevented the p-value=0.0833 from crossing the significance threshold.

The reviewer is correct in bringing up a possible discrepancy between this study and our previous result (Shaw et al., PNAS). However, this is most likely due to the different amounts of 20E injected in the two studies. In Shaw et al., injections of 2.5 ug (1x in the figure below) 20E resulted in a significant upregulation of *HPX15*, while a 10-fold dilution (0.25 ug) did not (Fig 6A of that manuscript; also attaching the figure below). In our current study we injected a reduced amount (0.63 ug 20E, therefore four times lower than the 1x concentration that had produced a significant upregulation of *HPX15*) as this is still in large excess relative to the 20E levels detected in the LRT after mating (Extended Table 1). Therefore, we can safely assume that in physiological conditions 20E does not significantly induce *HPX15*. We decided not to add this information in the discussion as it would considerably lengthen the text and it would not substantially change the manuscript.

Fig. 6A (Shaw et al., PNAS)

Line 172 - "Silencing of *Eck2* in virgin females caused a 3-fold increase in the relative ratio of 20E to 20E22P after blood feeding ". Level of knockdown was presented for dsEPP, but no data or comments on the level of knockdown for *Eck2* is presented. It needs to be included to support the statement on Line 172. Also, here in text the time scale after blood feeding is not given, though described as 26 h in the figure legend. This time should be added in text as well, particularly since this time point in a WT female corresponds to less 20E and more 20E22P.

We are grateful to the reviewer for pointing out this oversight. We have now added the *Eck2* knockdown efficiency as Extended Data Fig. 2b. We added "26 h" to the text.

Fig 5 and line 352. Data were pooled in all panels. Was this statistically appropriate to combine the data? For example, how were pooled % displayed if there was variation between replicates?

Combining the replicate data for Fisher's exact test was justified by the fact that all replicates showed the same trend. To conform to the highest statistical rigor while analyzing replicated contingency tables, in the revised manuscript we replaced Fisher's exact test of independence with Cochran–Mantel–Haenszel test for repeated tests of independence. The result/significance of the tests did not change, albeit it introduced some minor changes in p-values. We have uploaded the R script used to conduct the Cochran–Mantel–Haenszel test, which also contains our data from individual replicates (link included in the methods).

Reviewer Reports on the First Revision:

Referees' comments:

Referee #1 (Remarks to the Author):

I have reread the manuscript and the authors responses to all of the reviewers comments and it is my option that it is ready for publication. I think it is a very exciting and through piece of work that will be of general interest to those studying sex specific hormone function, and/or insect endocrinology.

Referee #2 (Remarks to the Author):

The author's responses and edits to the are satisfactory from my perspective. I've not further points.

Referee #3 (Remarks to the Author):

the authors have done a good job addressing my comments and those of other reviewers. The revised version of the manuscript is improved. I do not have any additional suggestions to make.

Author Rebuttals to First Revision:

Referees' comments:

Referee #1 (Remarks to the Author):

I have reread the manuscript and the authors responses to all of the reviewers comments and it is my option that it is ready for publication. I think it is a very exciting and through piece of work that will be of general interest to those studying sex specific hormone function, and/or insect endocrinology.

We thank the reviewer for these very positive comments.

Referee #2 (Remarks to the Author):

The author's responses and edits to the are satisfactory from my perspective. I've not further points.

We are glad that our responses were satisfactory to the reviewer.

Referee #3 (Remarks to the Author):

the authors have done a good job addressing my comments and those of other reviewers. The revised version of the manuscript is improved. I do not have any additional suggestions to make.

We are glad that we addressed the reviewer's comments and the revised manuscript is improved.